# Fibroblast state switching orchestrates dermal maturation and wound healing

Emanuel Rognoni[1,‡] (ID), Angela Oliveira Pisco[1,†,‡] (ID), Toru Hiratsuka[1] (ID), Kalle H Sipilä[1], Julio M Belmonte[2], Seyedeh Atefeh Mobasseri[1], Christina Philippeos[1] (ID), Rui Dilão[3] & Fiona M Watt[1,*] (ID)

## Abstract

Murine dermis contains functionally and spatially distinct fibroblast lineages that cease to proliferate in early postnatal life. Here, we propose a model in which a negative feedback loop between extracellular matrix (ECM) deposition and fibroblast proliferation determines dermal architecture. Virtual-tissue simulations of our model faithfully recapitulate dermal maturation, predicting a loss of spatial segregation of fibroblast lineages and dictating that fibroblast migration is only required for wound healing. To test this, we performed *in vivo* live imaging of dermal fibroblasts, which revealed that homeostatic tissue architecture is achieved without active cell migration. In contrast, both fibroblast proliferation and migration are key determinants of tissue repair following wounding. The results show that tissue-scale coordination is driven by the interdependence of cell proliferation and ECM deposition, paving the way for identifying new therapeutic strategies to enhance skin regeneration.

**Keywords** dermis development; fibroblast states; mathematical modelling; tissue architecture; wound healing

**Subject Categories** Development & Differentiation; Quantitative Biology & Dynamical Systems; Stem Cells

**Mol Syst Biol. (2018) 14: e8174**

## Introduction

Mammalian skin comprises two mutually dependent layers, the epidermis and the dermis, which form through highly coordinated epithelial–mesenchymal interactions during development (Fuchs & Horsley, 2008; Watt, 2014). At embryonic day 12.5 (E12.5), mouse epidermis comprises one or two cell layers, and the dermis appears homogeneous in composition. During development, the dermis evolves from a multi-potent pool of Pdgfrα[+] fibroblasts. These become lineage-restricted at embryonic day 16.5 (E16.5), such that Lrig1 expressing fibroblasts give rise to the upper (papillary) dermis, while Sca1/Dlk1-positive fibroblasts give rise to the lower (reticular) dermis and hypodermis (Driskell *et al*, 2013).

The papillary dermis is distinguishable from the reticular dermis because of its higher cellular density and relative paucity of fibrillar collagen. Functionally, the papillary lineage is required for hair follicle formation in skin reconstitution assays, whereas the lower lineage gives rise to the fibroblasts that mediate the initial phase of wound repair (Driskell *et al*, 2013; Rognoni *et al*, 2016). By postnatal day 2 (P2), the hypodermis has formed, comprising differentiated adipocytes and preadipocytes, while fibroblasts from the upper dermis differentiate into the hair follicle arrector pili muscle. By P10, fibroblasts stop proliferating and dermal expansion results in separation of clonally related fibroblasts (Rognoni *et al*, 2016). In addition, resident immune cells, neuronal cells and endothelial cells are recruited, giving rise to the adult dermis.

It has been shown recently that epidermal cells coordinate a tissue-scale behaviour during wound repair, whereby the epithelium organises directional migration and proliferation in overlapping regions oriented towards the wound (Aragona *et al*, 2017; Park *et al*, 2017). Upon wounding, dermal fibroblasts become activated, as evidenced by expression of α-smooth muscle actin (α-sma), start proliferating, migrate to the wound and deposit ECM, reconstituting the wound bed (Eming *et al*, 2014; Shaw & Martin, 2016). The upper and lower lineages enter the wound with different kinetics (Driskell *et al*, 2013; Rognoni *et al*, 2016), and adipocytes can be replenished from α-sma[+] fibroblasts (Plikus *et al*, 2017). However, how the growth and spatial organisation of dermal fibroblast subpopulations is regulated is currently unknown.

Here, we elucidate how the tissue-scale coordination of fibroblast behaviour is achieved during dermal development and homeostasis. Using a combination of cell biology techniques and mathematical modelling, we were able to demonstrate that fibroblast behaviour switching between two distinct states—proliferating and depositing ECM—is necessary and sufficient to define dermal architecture. These cellular states are balanced by a negative feedback loop between ECM deposition/remodelling and proliferation. Understanding the interdependence of cell behaviour and ECM is

1 Centre for Stem Cells and Regenerative Medicine, King's College London, London, UK
2 Developmental Biology Unit and Cell Biology and Biophysics Unit, European Molecular Biology Laboratory, Heidelberg, Germany
3 Nonlinear Dynamics Group, Instituto Superior Técnico, Lisbon, Portugal
*Corresponding author. Tel: +44 207188 5608; E-mail: fiona.watt@kcl.ac.uk
‡These authors contributed equally to this work
†Present address: Chan Zuckerberg Biohub, San Francisco, CA, USA

potentially important for identifying new therapeutic strategies to enhance skin regeneration.

# Results

## Dermal maturation is driven by an inverse correlation between fibroblast proliferation and ECM deposition

By combining fibroblast density measurements with dermis volume calculations (Rognoni *et al*, 2016), we estimated the number of cell divisions during embryonic (E17.5 to P2) and post-natal (P2 to P50) growth (Figs 1A and EV1A, Table EV1). Our data indicated that postnatally the dermis volume increased approximately 13 fold with minimal proliferation, as our model only predicts 1.3 cell divisions. In contrast, during embryonic development, the dermis volume increased proportionally to the change in cell number, indicating that at this stage tissue growth is driven by cell proliferation.

In line with this observation, we found that most fibroblasts are proliferating (Ki67[+]) at E10.5, but with age they progressively arrest in the G1 cell cycle phase, which we define as quiescence (Basak *et al*, 2017), without undergoing apoptosis (Rognoni *et al*, 2016; Figs 1B upper panel and C, and EV1B). The entry into quiescence coincided with a sharp increase in collagen deposition (quantified by Picrosirius red staining; Fig 1B lower panel). Quantitation of changes in proliferation and ECM deposition suggested an inverse correlation over time (Fig 1D), which led us to hypothesise that dermal growth consists of two phases. During the initial phase, tissue expansion is due to proliferation, as the ratio of dermis volume to cell number is approximately 1:1. The second phase, corresponding to postnatal growth, is strongly associated with ECM deposition and remodelling, as between P2 and P50 the ratio of dermis volume to cell number is 4:1. These observations are supported by publicly available microarray data for neonatal and adult back skin fibroblasts, which show that with age there is a reduction in genes associated with proliferation, together with an enrichment for GO terms for ECM

production and remodelling (Fig EV1C; Collins *et al*, 2011; Rognoni *et al*, 2016).

We further investigated the changes in the ECM by labelling with a collagen hybridising peptide probe (CHP), which recognises the triple helix structure of immature and remodelling collagen fibres (Fig EV2A, Hwang *et al*, 2017). Collagen fibre bundles were first detectable at E18.5 in the lower dermis, and in agreement with the Picrosirius red staining, further matured and expanded throughout the dermis with age. Moreover, ultrastructural analysis of P2 skin sections revealed that while there was no difference in collagen fibre diameter between upper dermis and lower dermis, collagen fibre bundle formation was evident in the lower dermis, whereas only small and dispersed collagen fibres were present in the upper dermis (Fig EV2B and C). At P2, the fibroblasts closest to the basement membrane were more proliferative than those in the lower dermis (Figs 1E and F, and EV2D). This indicates that the relationship between ECM deposition and fibroblast quiescence holds within different regions of the dermis at any given time point. We observed the same correlation when we examined human foetal and adult skin sections (Fig 1G–I). We therefore conclude that dermal fibroblasts exhibit differential growth behaviour at distinct developmental time points and dermal locations, either dominated by cell proliferation or ECM production.

## ECM negatively regulates fibroblast proliferation

To investigate whether the presence of ECM in a 3D environment is sufficient to prevent fibroblast proliferation, we isolated fibroblasts from P2 mice by flow cytometry and plated them on collagen-coated tissue culture plastic (TCP) or encased them at low density within collagen gels (Fig 2A). We found that culture on TCP resulted in a much greater increase in the number of fibroblasts during the recording period of 4 days than culture within collagen. The negative effect of the 3D collagen environment was reversible, as indicated by the fact that when cells were released by collagenase I treatment, previously non-proliferating fibroblasts resumed a highly proliferative rate soon after being re-plated on TCP (Fig 2B).

**Figure 1. Dermal architecture is defined by an inverse correlation between fibroblast proliferation and ECM deposition.**

A  Quantification of fibroblast density (number of PDGFRαH2BEGFP[+] cells, right; *n* = 3 biological replicates per time point) and dermis volume (left) with age (*n* = 7 for 12.5; *n* = 8 for 19.5, 25.5; *n* = 9 for 16.5, 17.5, 23.5; *n* = 10 for 29.5, 69.5; *n* = 11 for 10.5; *n* = 12 for 18.5, 21.5, 40.5; *n* = 20 for 31.5; biological replicates).

B  Changes in fibroblast proliferation and collagen deposition during dermal development in mice. Immunofluorescence staining for Ki67 (red) of PDGFRαH2BEGFP (green) back skin at indicated developmental time points (upper panel). Polarised light images of Picrosirius red stained back skin sections shown in binary images at indicated time points (lower panel).

C  Percentage of PDGFRαH2BEGFP[+] cells in G1 cell cycle phase with age (*n* = 1 for 12.5; *n* = 2 for 21.5, 29.5; *n* = 3 for 16.5, 17, 19.5; *n* = 4 for 10.5; *n* = 5 for 17.5, 18.5, 69.5; *n* = 7 for 23.5; biological replicates) fitted using an exponential model.

D  Quantification of fibroblast proliferation (left; *n* = 4 for 16.5, 29.5; *n* = 3 for 10.5, 18.5, 21.5, 41.5, 59.5, 69.5; *n* = 2 for 17.5, 19.5; biological replicates) and collagen density (right; *n* = 4 for 16.5, 21.5, 24.5, 29.5, 33.5, 35.5, 41.5, 73.5; *n* = 3 for 10.5, 17.5, 18.5, 19.5, 31.5, 38.5; biological replicates) with age. Error bars represent standard deviation of the biological replicates.

E  Immunofluorescence image of R26Fucci2a x Dermo1Cre back skin at P2 where mVenus-hGem (green) expressing cells are in S/G2/M phase, and mCherry-hCdt1 (red)-positive cells are in G1 phase.

F  Percentage of labelled cells in S/G2/M cell cycle phase in the upper and lower dermis at P0 (*n* = 3 biological replicates) and P2 (*n* = 4 biological replicates).

G–I  Changes in dermal cell proliferation and collagen deposition in human skin. (G) Immunofluorescence staining for vimentin (red) and Ki67 (green) of back skin at indicated time points (upper panel). Polarised light images of Picrosirius red stained back skin section shown as binary image at indicated time points (lower panel). (H) Quantification of proliferating cells in the upper and lower dermis (*n* = 3 biological replicates per time point). (I) Quantification of dermal cell proliferation (Ki67[+]) (left) and collagen density (right) with age (*n* = 3 biological replicates per time point). Note that cells in upper human dermis were more proliferative than cells in the lower dermis at all developmental time points analysed and that fibroblasts entered a quiescent, non-proliferative state before collagen was efficiently deposited.

Data information: Data shown are means ± s.d. Nuclei were labelled with DAPI (blue in B, E, G). Scale bars, 100 μm. wk, week.

    

To determine whether the negative effect of ECM on fibroblast proliferation held true in a more physiological model, we reconstituted decellularised human dermis (DED) with primary human fibroblasts and keratinocytes and cultured them at the air–medium interface, as described previously (Rikimaru *et al*, 1997; Philippeos *et al*, 2018; Fig 2C–G). After 3 weeks, all fibroblasts entered a quiescent state (Fig 2D and E), recapitulating the fibroblast behaviour observed in adult mouse skin (Fig 1B–D). However, when we injected collagenase mixture into the DED, the fibroblasts surrounding the injection sites started to proliferate within 48 h (Fig 2F and G). Thus, we conclude that in both *in vitro* assays the 3D ECM environment negatively regulates fibroblast proliferation. Nevertheless, the inhibition of proliferation is reversible and occurs in the presence or absence of keratinocytes.

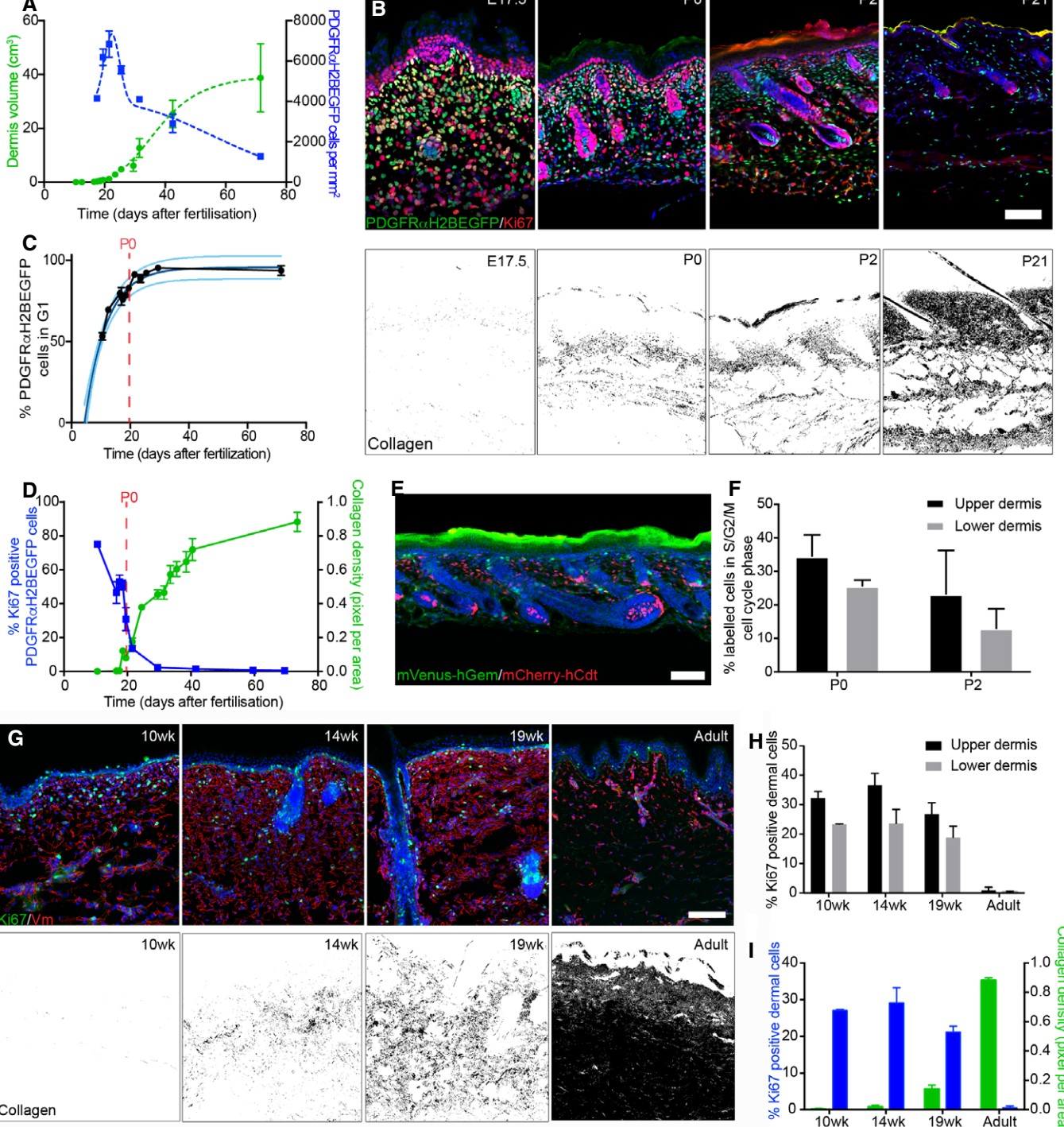

**Figure 1.**

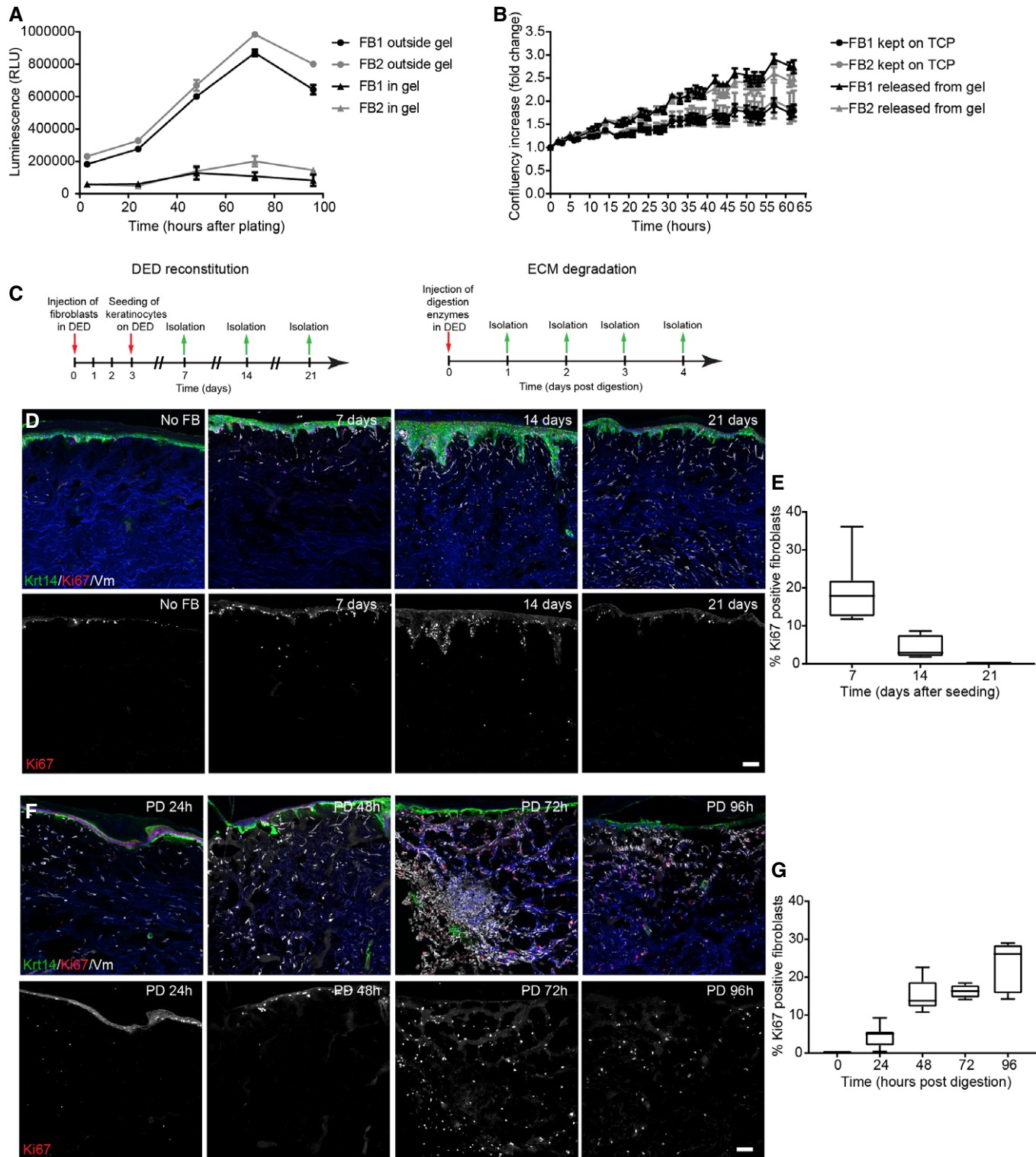

**Figure 2.**

### Modelling a switch between two fibroblast states

To create a mathematical model deconstructing the inverse correlation between proliferation and ECM production, we assumed that

fibroblasts switch between two states, proliferating fibroblasts (PF, with proliferating rate $\kappa_1$) and quiescent fibroblasts (QF), with transition rates $\kappa_2$ and $\kappa_{-2}$, respectively (Fig 3A; Materials and Methods). Following the experimental observations, we conjectured that

**Figure 2.  ECM triggers fibroblast state switching.**

A    Two independent primary mouse fibroblast isolations FB1 (black) and FB2 (grey) were cultured in (triangles) and outside (circles) collagen gels for up to 5 days. Cell number was measured by the CellTiter-Glo assay at indicated time points ($n$ = 3 technical replicates of one representative experiment repeated three times).

B    Two independent primary mouse fibroblast isolations FB1 (black) and FB2 (grey) were cultured in collagen gels for 3 days before being released by collagenase treatment and plated on TCP for 4 days (triangles). Controls were kept on TCP for 3 days before plating (circles). Fold increase in confluency 24 h after plating of collagen gel released cells and cells kept on TCP is shown ($n$ = 3 technical replicates).

C    Experimental design of decellularised dermal (DED) organotypic cultures for reconstitution (left panel) and ECM degradation (right panel).

D, E    Repopulation of DED organotypic cultures recapitulates adult dermal homeostasis. (D) Immunofluorescence staining for Krt14 (green), Ki67 (red) and vimentin (white) of DED organotypic cultures with and without fibroblasts at indicated time points after seeding. (E) Quantification of Ki67-positive fibroblasts at indicated time point after seeding ($n$ = 8 sections per condition of one representative experiment repeated twice).

F, G    ECM degradation in DED organotypic cultures by collagenase treatment. (F) Immunofluorescence staining for Krt14 (green), Ki67 (red) and vimentin (white) of DED organotypic cultures at indicated time points after collagenase treatment. (G) Quantification of Ki67-positive fibroblasts at indicated time points after collagenase treatment ($n$ = 8 sections per condition of one representative experiment repeated twice). Note that fibroblasts around the injection site start to proliferate after 48 h.

Data information: Nuclei were labelled with DAPI (blue) (D, F). Scale bars, 100 μm (D, F). Data shown are means ± s.d. Fb, fibroblasts; PD, postdigestion; TCP, tissue culture plastic; h, hours. Boxplot horizontal bar represent the median, box limits the 25% percentile and the whiskers the min/max (E, G).

the existence of ECM would negatively regulate PF ($\kappa_4$), pushing the equilibrium towards a state where PF were minimal and both QF and ECM deposition/remodelling were maximal. The derived ordinary differential equation (ODE) model is shown in Fig 3B. To fit the experimental data, we defined our multi-objective optimisation problem adapted to the particularity of having two data sets to fit (PF and ECM, Fig 1D) and followed a Monte Carlo technique to find the solutions (Fig 3C; Dilão et al, 2009; Dilão & Muraro, 2010; Dilão & Sainhas, 2011; Muraro & Dilão, 2013). For the parameters obtained, our model had one stable state for which all variables (PF, QF and ECM) were non-negative: that corresponded to the adult dermis. We conclude that a simple model whereby ECM represses fibroblast proliferation via a negative feedback loop is the key element setting dermal architecture.

### A 3D tissue model recapitulates dermal maturation

In order to test the mechanism of fibroblast behaviour and spatial organisation during dermal maturation, we developed a 3D model in CompuCell3D (Belmonte et al, 2016; Hirashima et al, 2017; Swat et al, 2012). We conceptualised the whole animal as a cylindrical object and focused on modelling the dermis between neighbouring hair follicles, to avoid considering changes in the skin associated with the hair growth cycle (Donati et al, 2014). We initialised our body segment model as a simple cylindrical segment, where epithelial tissue (green) surrounded the proliferative fibroblasts (blue) enclosing a lumen representative of the inside of the animal body (white; Fig EV3A). To explore the fundamental dermal architecture, we focused solely on dermal fibroblast behaviour and excluded other cell types, such as immune, neuronal or endothelial cells.

Since fibroblasts close to the basement membrane were more proliferative (Figs 1E and F, and EV2D) and the switch in fibroblast behaviour also correlated with a decrease in basal keratinocyte proliferation (Fig EV3B), we proposed that the differential spatial proliferation of fibroblasts during development was influenced by an epidermal gradient (Fig EV3C, Collins et al, 2011; Lichtenberger et al, 2016). In agreement with the experimental evidence (Lichtenberger et al, 2016), epidermal signal strength was assumed to decline with age, accounting for the decline in proliferation. Therefore, in this model, the epidermal gradient directly impacted on fibroblast division capability. To account for the effect of ECM

on fibroblast proliferation, we linked the transition from the proliferative state to quiescence with the amount of ECM surrounding the cell.

While both proliferating and quiescent fibroblast populations can produce ECM, in line with the model in Fig 3A, quiescent cells deposited and remodelled ECM more efficiently. We included an adipocyte layer to represent the process of fibroblast differentiation into adipocytes in the deepest dermis, based on experimental data demonstrating the timing of appearance of the dermal white adipose tissue (DWAT; Rognoni et al, 2016); however, further modelling of DWAT maturation during development was not considered.

The fundamental tissue-scale behaviour of the dermis was fully recapitulated in our spatial model (Fig 4A–C, Movie EV1). We started with a pool of proliferative fibroblasts that respond to the presence of the epidermal gradient [Monte Carlo step (MCS) 0, ~E10.5]. When the proliferative cells become surrounded by ECM, the cells stop dividing and deposit ECM more efficiently. The cells in the lower dermis differentiate into adipocytes (MCS > 300) in a spatially controlled manner, that is, only cells in direct contact with the body wall are able to do so. At the first step of dermal maturation, we went through a phase of homogeneous tissue (MCS < 100), where we had minimal ECM and all cells were proliferative. Next, proliferating cells switched to a quiescent state, increasing ECM deposition, and the proliferative rim converged to the region around the epidermis (MCS ~200, ~E17.5). After MCS > 300 (~P0), cells close to the lumen differentiated into adipocytes, forming the hypodermis layer. After MCS > 450 (~P5), all proliferation ceased and the dermis started expanding due to the accumulation of ECM. As a reference, the model progression was optimised such that MCS = 300 coincided with the in vivo proliferation data measured at P0 (for full details please refer to the Materials and Methods section).

### Fibroblast organisation occurs without active cell migration during dermal maturation

We made two predictions from the computational model: dermal organisation is achieved without fibroblast migration and there is no spatial segregation of fibroblast lineages in adult dermis. To examine whether or not there is fibroblast movement within adult dermis, we performed in vivo live imaging of the back skin of adult PDGFRαH2BEGFP transgenic mice. We recorded the same field of

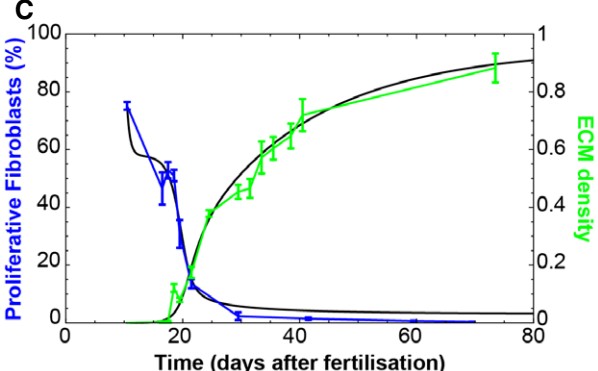

**Figure 3. A negative feedback loop between ECM and proliferation determines fibroblast state switching.**

A Schematic representation of the assumed relationships between proliferating fibroblasts (PF), quiescent fibroblasts (QF) and extracellular matrix (ECM) during dermal maturation. PF self-replicate with rate $\kappa_1$. PF and QF can interconvert with rates $\kappa_2$ and $\kappa_{-2}$, respectively. QF deposit ECM at rate $\kappa_3$. ECM promotes the switch from PF to QF at rate $\kappa_4$.

B System of ordinary differential equations (ODE) describing the process of dermal maturation. Because the system is dynamic, all entities can decay over time ($\kappa_6$ is the degradation rate of QF, $\kappa_7$ is the degradation rate of ECM, $\kappa_5$ is the degradation rates of PF and is incorporated into $\beta$ as $\beta = \kappa_1 GF - \kappa_2 - \kappa_5$ where GF stands for the growth factors we assume exist).

C Resulting simulation (black lines) of the system of equations in (B) and experimental data in Fig 1D. Error bars represent standard deviation of the biological replicates.

cells continuously for up to 700 min and detected only minimal cell movement (Fig 4D, Movie EV2), consistent with the prediction of the computational model.

In P2 dermis, there is clear spatial segregation of the papillary and reticular dermis (Driskell *et al*, 2013). However, without explicitly modelling active fibroblast migration, *in silico* lineage tracing indicated that fibroblasts lying closest to the epidermis would disperse throughout the dermis over time and eventually start contributing to the adipocyte layer (Fig 4E and F). Lower dermal fibroblasts would also disperse but rarely move into the upper

dermis (Fig 4G). The computational simulations were consistent with *in vivo* lineage tracing data. Fibroblasts from the upper dermis (labelled with Blimp1Cre or Lrig1CreER) became dispersed throughout the dermis, whereas lower dermal fibroblasts (labelled with Dlk1CreER) were predominantly confined to the lower dermis and DWAT (Fig EV3D and E; Driskell *et al*, 2013; Lichtenberger *et al*, 2016; Rognoni *et al*, 2016).

Together, our experiments and simulations demonstrated that no active cell migration was required to generate the architecture of the mature dermis. Instead, our simulations suggested that dermal fibroblasts become gradually displaced over time as a consequence of the switching between the two states (proliferative versus ECM synthesising), driving tissue expansion.

### The proliferation–ECM-negative feedback loop restores tissue homeostasis following wound healing

Our computational model indicates a unidirectional shift from the proliferating state to the state of ECM production during dermal maturation. However, it is well established that upon wounding dermal fibroblasts become activated and re-start proliferating. How these changes are coordinated on the tissue scale, such that the regular architecture is re-established, is unknown.

To test fibroblast state reversibility, we created 2-mm-diameter circular full-thickness wounds on adult mouse back skin and analysed them at different locations and time points (Fig 5A and B). Upon wounding, fibroblasts close to the wound became rapidly activated and expression of α-sma peaked at postwounding day 7 (PW7; Figs 5C and D, and EV4A). We observed high fibroblast proliferation in the upper and lower wound bed during the early phases of wound repair, suggesting that fibroblasts in different locations exit their quiescent state with identical probabilities (Figs 5E and F, and EV4B). By PW7, fibroblasts reached their maximal cell density in the wound bed centre, exceeding the density outside the wound (Fig 5G). This coincided with the first appearance of collagen fibres with random orientations in the upper and lower wound bed (Figs 5H and EV4C). During the advanced wound healing phase (> PW10), most fibroblasts stopped proliferating (Fig 5F and I). At PW14, fibroblasts in the wound bed centre were still activated and had a higher density than in the surrounding tissue, while collagen fibres were less mature (Figs 5C–H and EV4C). In the late wound healing phase (> PW21), keratinocyte proliferation at the wound site remained high, while fibroblast density was restored to normal (Fig 5I).

When comparing fibroblast proliferation with ECM accumulation in the wound bed, we observed that the inverse correlation was recapitulated inside the injured area (Fig 6A). Outside the wound, there were only minor changes in collagen or fibroblast density (Fig 6B), indicating that only fibroblasts close to the wound site were actively involved in the repair process. To determine whether the switch from quiescence back to proliferation explained the observed kinetics of wound healing, we revisited the mathematical model in Fig 3. As the experimental data points towards a *de-novo* accumulation of cells inside the wound bed, the model was expanded with the additional assumption that proliferating (PF) and quiescent fibroblasts (QF) at the boundary of the wound could move. Under these conditions, the spatial structure of the dermis was described by the partial differential equation model in Fig 6C,

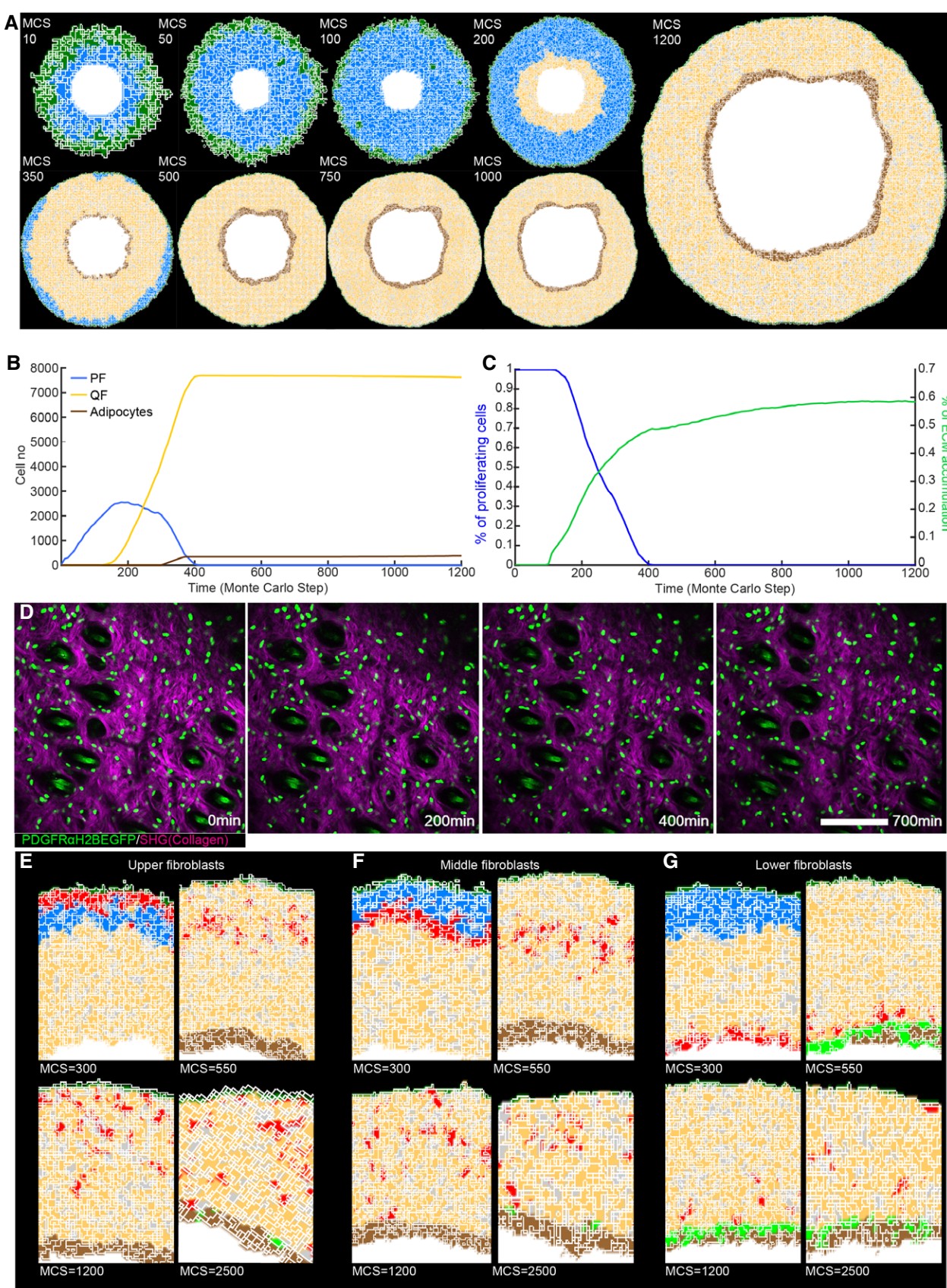

**Figure 4.**

**Figure 4. Development of a 3D tissue model and *in vivo* live imaging during dermal maturation.**

A    (*x,y*) Cross-section of the computational model simulation of the dermal maturation process at the indicated Monte Carlo step (MCS). Colour code indicates epidermis (green), proliferating fibroblasts (blue), quiescent fibroblasts (yellow), ECM (grey) and adipocytes (brown) (see accompanying Movie EV1).

B    Abundance of the cell populations during the course of the simulation.

C    Quantification of fibroblast proliferation and ECM density in the simulation.

D    Live imaging of adult PDGFRαH2BEGFP mouse back skin. Representative time-lapse images of adult PDGFRαH2BEGFP (green) dermis with collagen shown as second harmonic generation (SHG) in purple at indicated imaging time points. Scale bar, 100 μm (see accompanying Movie EV2).

E–G    *In silico* lineage tracing of fibroblasts. (E) Labelling of fibroblasts close to the epidermis at indicated MCS. (F) Labelling of fibroblasts in the middle of the dermis at indicated MCS. (G) Labelling of fibroblasts close to the inside of the body at indicated MCS.

where *D*, the diffusion coefficient, represented cell motility. The qualitative behaviour of our mathematical model corroborated the experimental data. Accordingly, there was an increase in proliferation during the initial repair phase (4 days after wounding; Fig 6D). Our mathematical model thus showed that the introduction of a diffusion coefficient was necessary and sufficient to allow the proliferation/ECM-negative feedback loop to describe the fibroblast behaviour observed during wound healing (Fig 6E, Movie EV3).

### Active cell migration is essential for efficient wound repair

We next revisited our spatial model to investigate the kinetics of fibroblast proliferation, migration and ECM deposition during wound healing. We simulated a 2 mm wound on adult mice (MCS 1,200, ~P50; Fig 7A) by removing a tissue slice and associated gradients and filling the space with a blood clot (red cells in the simulation) and infiltrating immune cells (purple cells in the simulation). These cells added a local wound healing gradient that decayed over time postwounding, mimicking the pro-proliferative effect of the blood clot and infiltrating immune cells on the surrounding fibroblasts *in vivo* (Fig EV5A; Eming *et al*, 2014). We modelled the fibroblast response to tissue wounding disregarding cell position in the tissue. Accordingly, fibroblasts in the vicinity of the wound became activated (cells in pink in the simulation) in response to the local and temporally restricted wound healing gradient and re-entered the high proliferation state. We introduced cell migration in the form of movement towards the wound healing gradient for the cells that were in the vicinity of the wound, enabling activated fibroblasts to invade the wound bed. The epidermis also proliferated in response to the wound healing gradient, which promoted the migration of proliferating keratinocytes to close the wound.

With this expanded model, our simulations were able to recapitulate the steps of the wound healing process and dissect the distinct fibroblast state/behaviour changes (Fig 7A, Movie EV4). Upon wounding, fibroblasts became activated and accumulated at the wound edge. While epidermal cells closed the wound and isolated the blood clot, activated fibroblasts invaded the space occupied by the immune cells and displaced them, before depositing and remodelling the ECM. At the later stage, during wound resolution, the cells around the wound area entered quiescence, restoring normal tissue physiology.

Since our models predicted that active cell migration is necessary for dermal repair and our spatial simulation corroborated the finding, we investigated fibroblast migration kinetics *in vivo*. As before, we created 2-mm-diameter circular wounds on adult PDGFRαH2BEGFP back skin and imaged them at the indicated time points (Fig 7B). Maximum projections of *Z*-stack images of the whole wound area revealed a sharp increase in fibroblast density at the wound borders 4 days postwounding (Fig 7C). No fibroblast migration was observed at PW2 (Fig EV5B, Movie EV5), whereas at PW4 we observed fibroblasts rapidly migrating into the wound bed (Figs 7D and EV6A, Movie EV6). Fibroblast migration stopped by PW7 (Fig EV6B), correlating with the appearance of collagen fibres in the wound bed (Fig EV4C). The lack of active migration at PW15, when fibroblast density in the wound bed was still high, suggested that fibroblasts were dispersed, while ECM was actively deposited and remodelled (Fig EV6C, Movie EV7). This was shown by the higher CHP signal in the wound bed even after PW21 (Fig EV4C).

To investigate whether fibroblasts from different locations expanded with different probabilities during wound repair, we performed lineage tracing experiments by crossing Confetti reporter mice either with Blimp1Cre mice, labelling fibroblasts in the upper dermis, or with Dlk1CreER mice, labelling fibroblasts in the lower dermis (Fig 7E; Driskell *et al*, 2013). When analysing the clonal distribution 10 days postwounding, we observed that clones of the upper and lower dermis were able to expand and redistribute in the entire wound bed, independent of their initial location (Fig 7F–H). In summary, the tissue-scale model, *in vivo* imaging and lineage tracing revealed that fibroblasts of the upper and lower dermis accumulate at the wound edges and migrate into the wound bed, before expanding and dispersing through the entire wound bed.

## Discussion

In this study, we demonstrated that mouse dermal expansion is driven by two sequential growth phases—pre- and postnatal—each dependent on cellular state switching. Using a combination of *in vivo* and *in silico* techniques, we showed that dermal architecture is established by a negative feedback loop between the deposition/remodelling of ECM and fibroblast proliferation. The temporally regulated switch from proliferation to quiescence is sufficient to achieve the tissue-scale coordination of fibroblast behaviour observed during dermal development and homeostasis. Together, our experimental data and validated tissue simulations support a model whereby fibroblast migration is a critical discriminator between dermal development and wound healing.

During both dermal maturation and repair, tissue architecture is established via a switch between fibroblast proliferation and ECM deposition. Proliferating fibroblasts progressively enter a quiescent state for efficient ECM production and remodelling. The presence of ECM promotes the cellular switch from the proliferative state towards the quiescent state and leads to cell separation via ECM-mediated displacement (Fig 8—Development). As a general principle, coordinated tissue growth and homeostasis require many cell

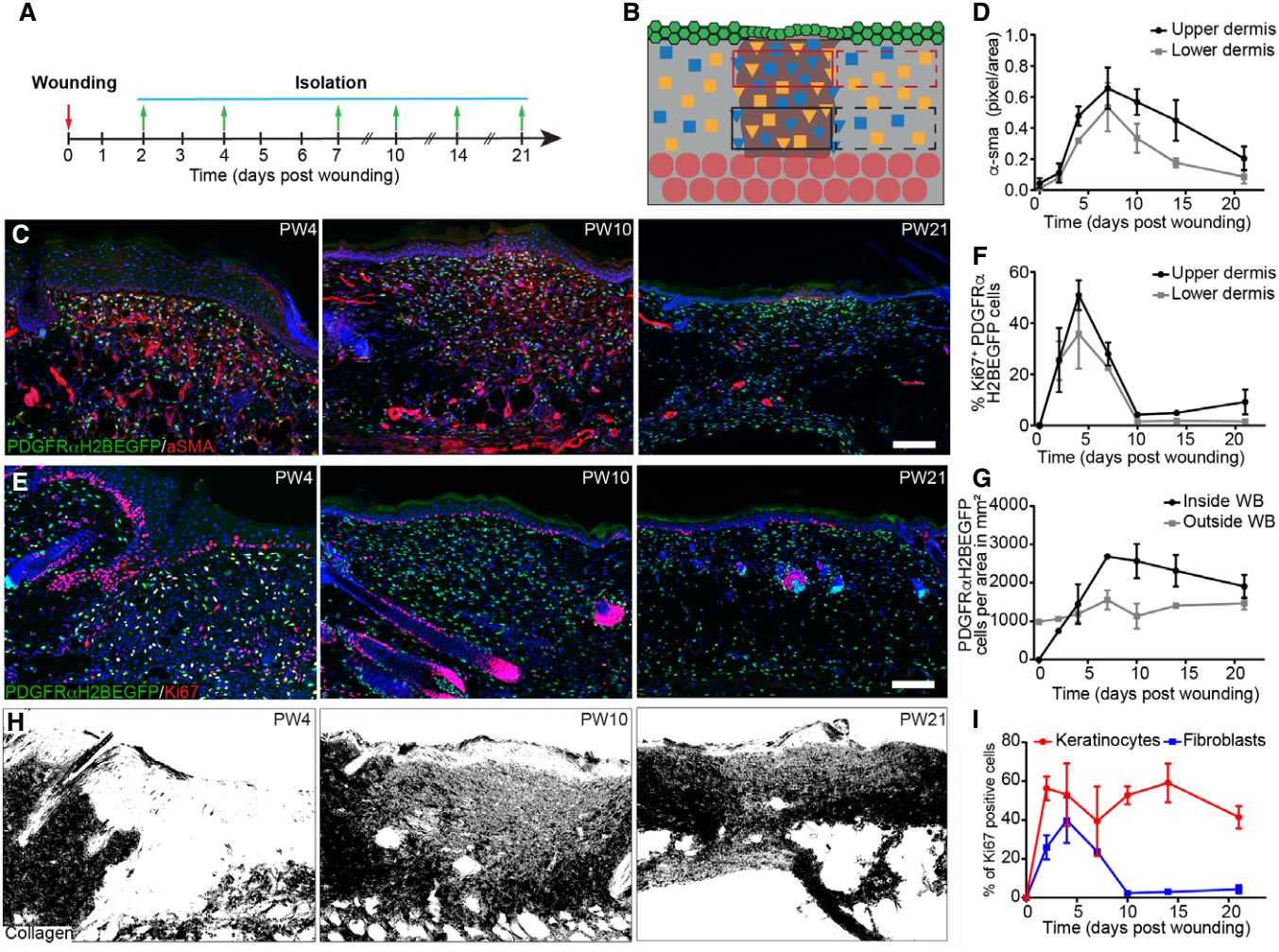

**Figure 5.  Fibroblast proliferation and ECM changes during wound healing.**

A, B  Experimental design of wound healing time course analysis. (A) Wound healing time course. Skin was collected at the indicated time points. (B) Quantification strategy is schematically illustrated. Red boxes upper dermis; black boxes lower dermis; solid lines inside and dashed lines outside the wound site.

C, D  Fibroblast activation during wound repair. Immunofluorescence staining for α-sma (red) of PDGFRαH2BEGFP (green) wound sites at indicated postwounding (PW) times (C) and quantification of α-sma fluorescence intensity in pixels per area inside the upper and lower wound bed (*n* = 4 PW0, PW2, PW7, PW21; *n* = 3 PW10, PW14; *n* = 2 PW4 biological replicates) (D).

E–H  Changes in fibroblast density, proliferation and collagen deposition during wound repair. (E) Immunostaining for Ki67 (red) of PDGFRαH2BEGFP (green) back skin sections at indicated wound healing time points. (F) Quantification of fibroblast proliferation in the upper and lower dermis inside the wound bed over time (*n* = 4 PW0, PW2, PW21; *n* = 3 PW4, PW10; *n* = 2 PW7, PW14 biological replicates). (G) Quantification of fibroblast density during wound repair inside and outside the wound bed (*n* = 4 PW0, PW2, PW21; *n* = 3 PW4, PW10; *n* = 2 PW7, PW14 biological replicates). (H) Polarised light images of Picrosirius red stained back skin section shown as binary image at indicated time points after wounding.

I  Quantification of proliferating (Ki67-positive) keratinocytes and fibroblasts during wound repair over time (*n* = 4 PW0, PW2, PW21; *n* = 3 PW4, PW10; *n* = 2 PW7, PW14 biological replicates).

Data information: Nuclei were labelled with DAPI (blue) (C, E). Scale bars, 100 μm (C, E). WB, wound bed. Error bars represent standard deviation of the biological replicates.

types, including stem cells and fibroblasts, to stop dividing and acquire a quiescent state (Cheung & Rando, 2013). This state is defined as a controlled and reversible cell cycle arrest, which can last for years. In cell culture, human diploid fibroblasts can enter quiescence in response to signals such as contact inhibition, loss of adhesion or starvation, which are not present *in vivo* (Coller *et al*, 2006). Interestingly, these different quiescence stimuli induce distinct gene expression signatures, indicating that quiescence is the manifestation of a collection of states rather than a common cell cycle arrest state (Wolf *et al*, 2009; Sang & Coller, 2009). Furthermore, cellular quiescence is an actively maintained state, not solely a consequence of cell cycle arrest, as neither CDK inhibition nor overexpression of CDKI induces a quiescence-specific gene expression program (Coller *et al*, 2006; Sang & Coller, 2009).

Many functional changes occur during induction of cellular quiescence, including changes in metabolism, autophagy, gene expression and chromatin structure (Lemons *et al*, 2010). Surprisingly, besides actively reinforcing the non-dividing cell cycle state

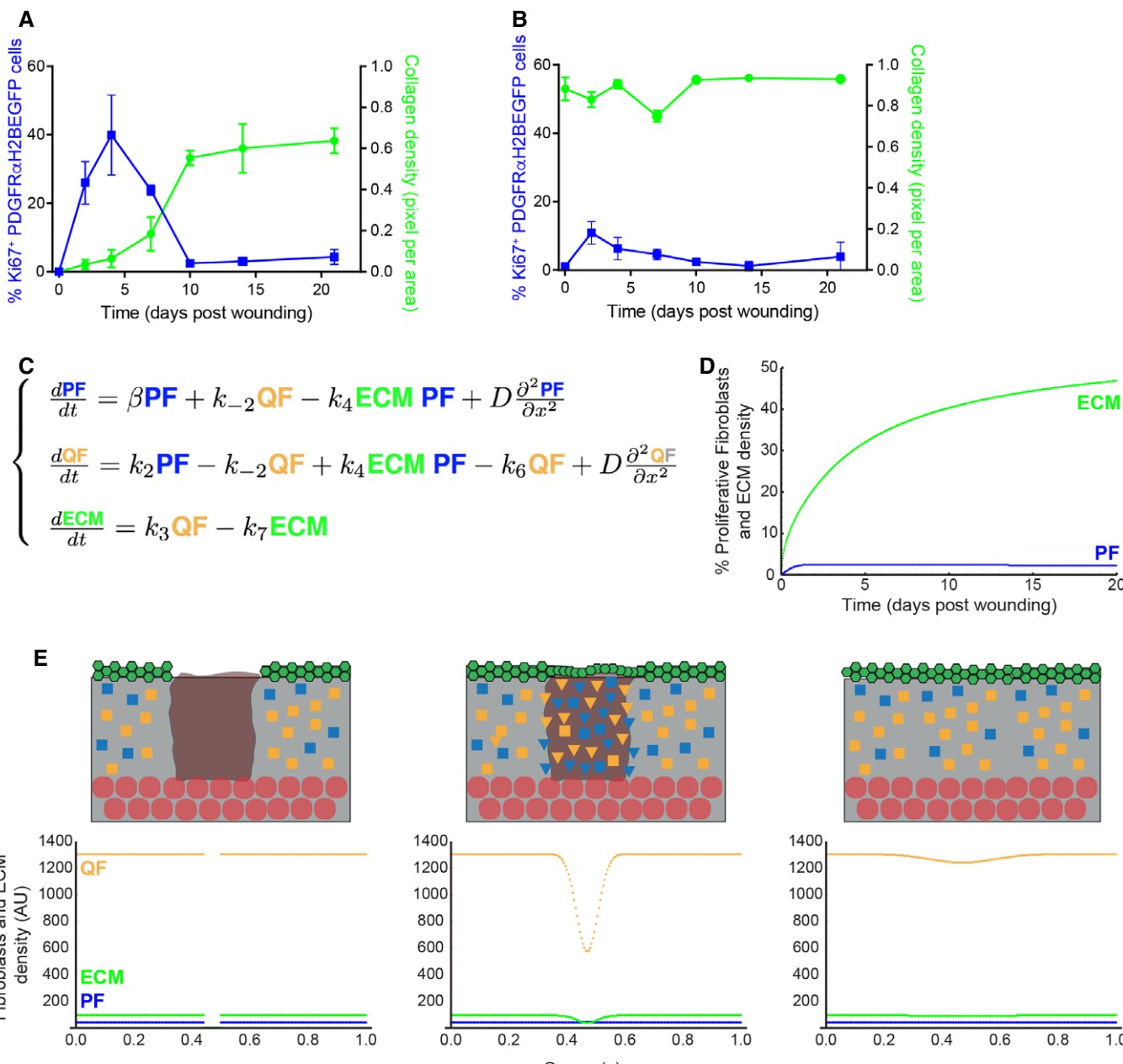

**Figure 6.   The negative feedback loop between ECM and proliferation is necessary and sufficient to restore tissue architecture.**

A, B   Quantification of fibroblast proliferation (left; *n* = 4 PW0, PW2, PW21; *n* = 3 PW4, PW10; *n* = 2 PW7, PW14 biological replicates) and collagen density (right; *n* = 4 PW0, PW2; *n* = 3 PW4, PW7, PW10, PW14; *n* = 2 PW21 biological replicates) inside (A) and outside (B) the wound bed over time. Error bars represent standard deviation of the biological replicates.

C   System of partial differential equations describing the process of dermal wound healing. The system in Fig 3B was expanded to incorporate the diffusion coefficient *D* with the associated spatial term.

D   Temporal evolution of proliferating fibroblasts (PF) and dermal ECM density over time. The model is able to describe the qualitative behaviour observed in panel (A).

E   Simulation of dermal wound closure showing the distribution of PF (blue), QF (yellow) and ECM (green) at the time of wounding and at two subsequent time points. From left to right, we performed a wound by imposing that both proliferating cells and ECM are zero in a specific region and simulate the wound healing process during time with the model equations in panel (C). Note that the system re-establishes the original densities, corroborating the experimental evidence that only the region directly associated with the wound was affected. Please refer to associated Movie EV3. In the schemes above: ECM, grey; wound bed, brown; epidermis, green; adipocytes, round red circles; QF, yellow squares; PF, blue squares; activated fibroblasts, triangles.

and repressing the transition into senescence or terminal differentiation, quiescent fibroblasts remain highly metabolically active. They increase expression of ECM proteins such as Col I and III, which is partly due to changes in expression of miRNAs such as miR-29, and in line with these findings, the quiescence gene signature is enriched in GO terms for ECM proteins (Suh *et al*, 2012). While the processes

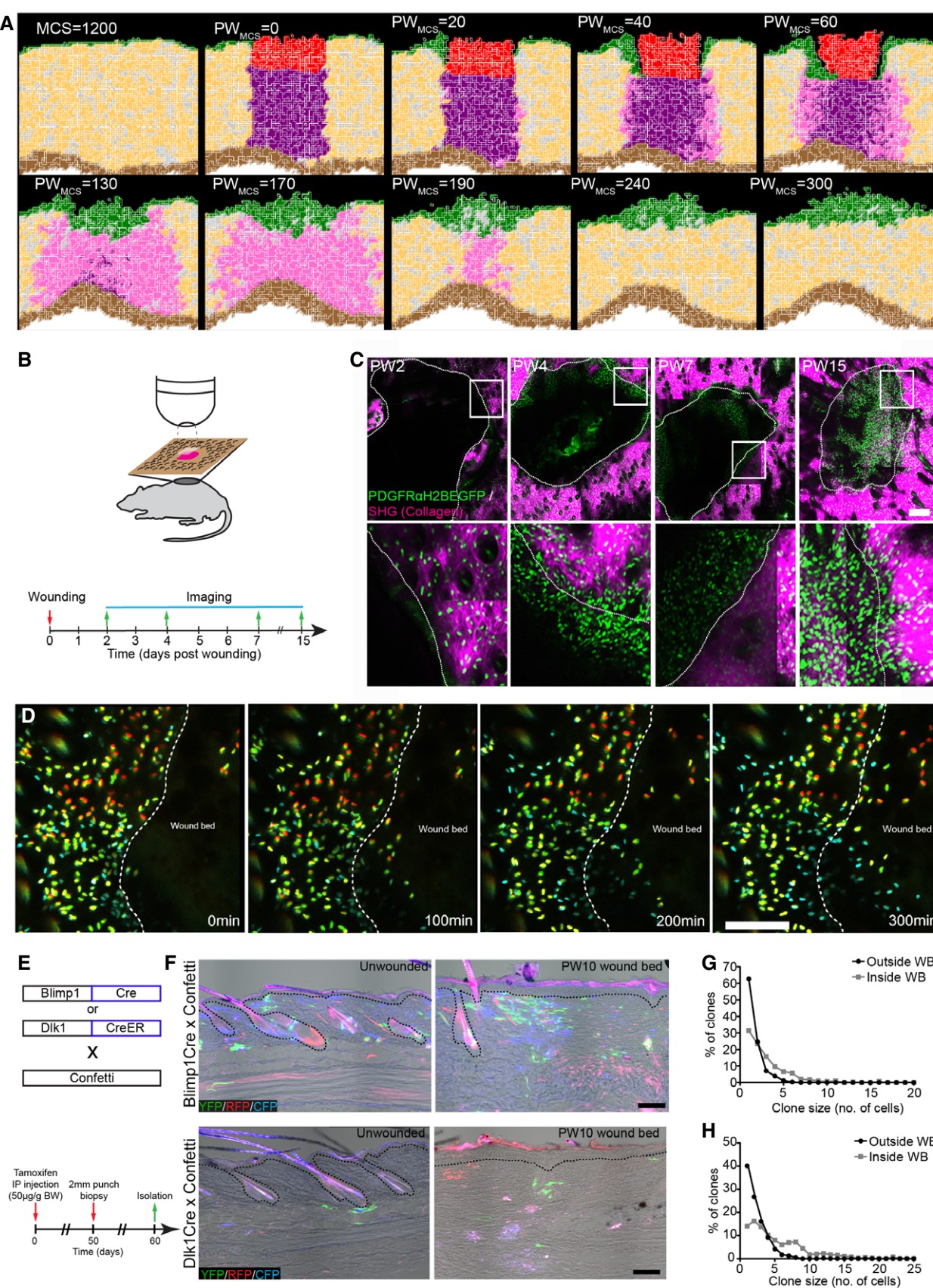

**Figure 7.**

**Figure 7.    3D model and *in vivo* live imaging of dermal wound repair.**

A        (*x,y*) Cross-section of the computational simulation of wound healing in adult dermis. Please refer to associated Movie EV4.

B–D    Live imaging of adult PDGFRαH2BEGFP back skin during wound healing. (B) Experimental design of *in vivo* imaging of wound healing. (C) Representative wound bed *Z*-stack stitched maximum projections of the entire wound bed (upper panel) showing fibroblasts (green) and collagen with second harmonic generation (SHG) in purple at indicated time points after wounding and before live imaging. Dotted line indicates wound edge, and boxed area indicates the position of the magnified area below. (D) Representative time-lapse images of adult wound bed at PW4, where *Z*-directional movement is visualised by pseudo-colouring the fibroblast nuclei (red, movement towards the basement membrane; blue, movement away from the basement membrane). Dotted line indicates wound edge. Please refer to associated Movie EV6 which shows fibroblasts actively migrating into the wound bed.

E–H    *In vivo* lineage tracing of upper (Blimp1Cre) and lower (Dlk1CreER) dermal fibroblasts during wound healing. (E) Experimental strategy for *in vivo* lineage tracing. (F) Immunofluorescence image of Blimp1Cre × Confetti (upper panels) and Dlk1CreER × Confetti (lower panel) back skin inside and outside the wound bed at 10 days postwounding. Colours show YFP (green), RFP (red) and CFP (blue) with bright field image overlaid. (G) Clone size quantification of Blimp1Cre-labelled fibroblasts inside and outside the wound bed at 10 days postwounding (*n* = 400 clones of four biological replicates). (H) Clone size quantification of Dlk1CreER-labelled fibroblasts inside and outside the wound bed at 10 days postwounding (*n* = 300 clones of two biological replicates).

Data information: Scale bars, 500 μm (C); 100 μm (D, F).

that drive fibroblast quiescence *in vitro* have been relatively well described, the signalling networks that stimulate and maintain quiescence of dermal fibroblasts *in vivo* are still unknown. Moreover, it is not known whether distinct fibroblast subpopulations exhibit different quiescence regulation mechanisms. Nevertheless, our microarray data suggest a role for chromatin, given that sorted fibroblasts at different ages show distinct signatures for GO terms such as chromatin assembly, DNA packaging and protein-DNA complex organisation. This is compatible with the observed fast and reversible fate switching from quiescence to proliferation during wound healing.

Before wounding, adult mouse has a steady-state distribution of fibroblasts in a quiescent state and associated ECM deposition (Fig 8—Homeostasis). Upon wounding, there is a local change in the concentration of cells and ECM and the system is no longer at steady state. In response to this, at very early stages of wound repair, fibroblasts actively migrate towards the wound bed and switch back to a proliferative state, a behaviour that is not restricted to a specific dermal layer (Fig 8—Tissue injury). Our integrative approach revealed that, in contrast to the observed behaviour during dermal development, both proliferation and migration of fibroblasts were essential in the early wound healing phase. At later stages, fibroblasts in the wound bed recapitulated the dermal maturation process by exiting the cell cycle, allowing for efficient ECM deposition and remodelling. To restore normal cell density, the wound bed fibroblasts were gradually dispersed by the newly synthesised and remodelled ECM, without the requirement for active cell migration. This sequence of changes in fibroblast behaviour during wound healing suggests that migration, proliferation and ECM synthesis are coordinated processes leading to fibroblast reorganisation.

Our proposed mathematical model demonstrates that dermal homeostasis can be achieved by a repressive mechanism in which the deposition of ECM promotes fibroblasts to stop proliferating. The resulting negative interaction is justified by the observation that the increase in collagen concentration is accompanied by the

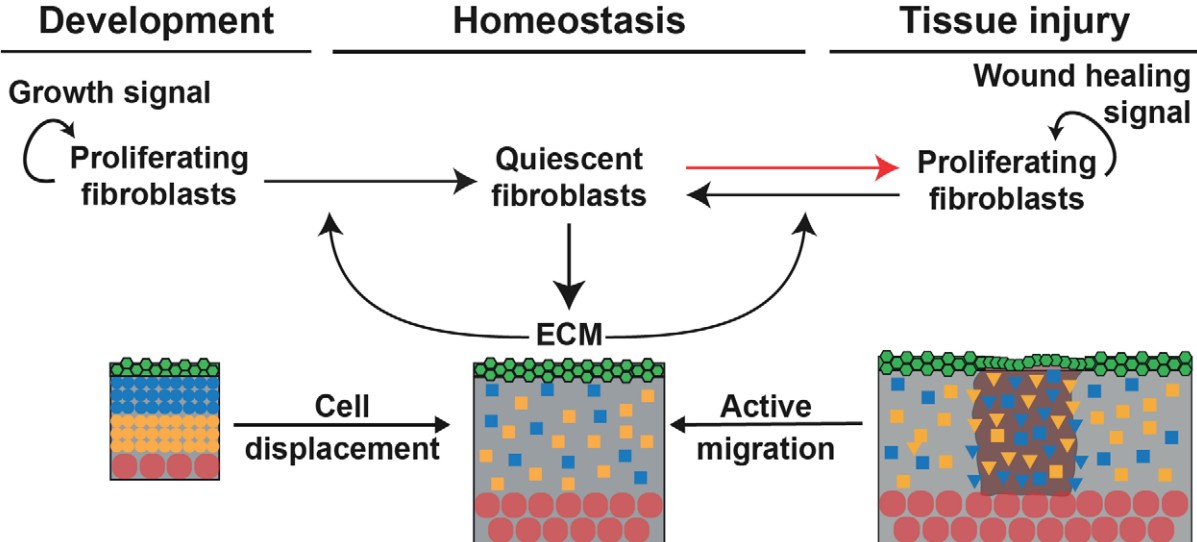

**Figure 8.    Schematic of how fibroblast state switching determines tissue architecture.**

During development, the tissue-scale behaviour of dermal fibroblasts is achieved and maintained by a feedback loop between ECM deposition and fibroblast switching from proliferating to quiescent. Upon wounding, quiescent cells re-enter the cell cycle, and tissue regeneration occurs via the sequential coordination of proliferation, active migration and cell displacement in the wound bed.

decrease in PF. On the other hand, the depletion of ECM during wounding, leading to regeneration, shows that when the repressor is absent, fibroblasts resume proliferation. To achieve regeneration in the wound, the simplest mechanism is to allow the motility of fibroblasts. This motility could be induced by the imbalance of fibroblast densities near the wound, which is a diffusive process. Thus, the mechanism whereby ECM deposition blocks proliferation holds true for tissue perturbation and defines the key element for re-establishing tissue architecture after wounding.

In skin, pathologies linked to quiescence can range from fibrosis, where there is excessive fibroblast activity following injury, or chronic wounds, where quiescent fibroblasts fail to re-enter the cell cycle and coordinate the wound healing response. Furthermore, tumour and tumour stroma cells seem to adopt cellular quiescence mechanisms to ensure long-term survival and escape chemotherapy-mediated killing (van Deursen, 2014). By developing a computational model, we were able to recapitulate the fundamental processes leading to tissue-scale behaviour during development, homeostasis and wound healing. Our model has enabled us to deconstruct at single cell level the events that occur during tissue regeneration, making it a promising tool to investigate responses to injury, disease and treatment strategies.

# Materials and Methods

### Transgenic mice and lineage tracing

All experimental procedures were carried out under the terms of a UK Home Office project license (PPL 70/8474) after local ethical review at King's College London. Animals were housed in IVC cages under standard conditions at the Biological Service Unit (BSU), King's College London. All mice were outbred on a C57BL6/CBA background, and male and female mice were used in experiments that included Blimp1-Cre (Robertson *et al*, 2007), Dlk1CreERt2 (Driskell *et al*, 2013), Lrig1CreER (Page *et al*, 2013), PDGFRαH2-BeGFP (Hamilton *et al*, 2003), ROSAfl-stopfl-tdTomato (Jackson Laboratories, 007905), CAGCATeGFP (Kawamoto *et al*, 2000), Dermo1Cre (Jackson Laboratories, 008712), R26Fucci2a (Mort *et al*, 2014) and Confetti mice (Snippert *et al*, 2010). Animals were sacrificed by CO₂ asphyxiation or cervical dislocation. No specific method for randomisation, blinding or estimation of sample size was used. All efforts were made to minimise suffering for mice.

For *in vivo* cell cycle analysis, R26Fucci2a mice were crossed with Dermo1Cre mice and back skin was analysed at the indicated time points. For lineage tracing, transgenic reporter mice were crossed with the indicated Cre or CreER lines. CreER was induced by injection with 10 μl tamoxifen (50 μg/g body weight; Sigma-Aldrich) intraperitoneally in newborn mice (P0), when DLK1 and Lrig1 are highly expressed in dermal fibroblasts (Driskell *et al*, 2013; Rognoni *et al*, 2016). After embryonic development, Blimp1 is not expressed in dermal fibroblasts, except for dermal papilla cells, enabling us to lineage tracing upper dermis (papillary) fibroblasts during development and wound healing (Donati *et al*, 2017; Driskell *et al*, 2013; Telerman *et al*, 2017). Tamoxifen for injection was dissolved in corn oil (5 mg/ml) by intermittent sonication at 37°C for 30 min. Tissue was collected at the indicated time points, briefly fixed with paraformaldehyde/PBS (10 min at room temperature),

embedded into optimal cutting temperature (OCT) compound and horizontal wholemounts or thin 12-μm-thick sections were prepared. Horizontal wholemount sections were mounted in glycerol and immediately imaged with a Nikon A1 confocal microscope using 10× or 20× objectives. Only YFP-, CFP- and RFP-labelled cells in Confetti mice were analysed, because the frequency of nuclear GFP-labelled cells was very low. All labelled cells of one colour within a 260 ± 50 μm radius (Driskell *et al*, 2013; Rognoni *et al*, 2016) were scored as being clonally related.

### Wound healing

Wound healing assays were performed as previously described (Rognoni *et al*, 2016). Briefly, analgesic EMLA cream (AstraZeneca) was applied topically to the back skin of adult mice 10 min before mice were anaesthetised using isoflurane (Cp-Pharma). A 2 mm punch biopsy (Stiefel) was used to make a full-thickness wound in the central back skin. Back skin was harvested at the time points indicated after wounding.

### Histology and microscopy

Human foetal tissue was obtained with appropriate ethical approval from the UK Human Developmental Biology Resource. Adult surgical waste skin from consenting patients undergoing plastic surgery was obtained from St George's University Hospitals NHS Foundation Trust, and the study was ethically approved by the National Research Ethics Service Committee UK (HTA Licence No: 12121, REC-No: 14/NS/1073). Mouse and human tissue samples were embedded in optimal cutting temperature compound (OCT, Life Technologies) prior to sectioning. For confocal imaging, cryosections of 12 or 25 μm thickness were fixed with 4% paraformaldehyde/PBS (10 min at room temperature), permeabilised with 0.1% Triton X-100/PBS (10 min at room temperature), blocked with 5% BSA/PBS (1 h at room temperature) and stained with the following primary antibodies to: vimentin (Cell Signaling, #5741), CD26 (R&D Systems, AF954), Sca1 (BD Pharmingen, clone E13-161.7), GFP (Abcam, ab13970 and A-11122, Thermo Fisher), Ki67 (Abcam, ab16667 and Invitrogen, clone SolA15), α-sma (Abcam, ab5694), Krt14 (BioLegend, 906001) and CD49f (BioLegend, clone GoH3). Samples were labelled overnight at 4°C, washed in PBS and labelled with secondary antibodies and 4,6-diamidino-2-phenylindole (DAPI; 1 μg/ml diluted 1:50,000; Thermo Fisher) for 1 h at room temperature. Samples were mounted with ProLong® Gold Antifade Mountant (Thermo Fisher). 60-μm horizontal wholemount sections were immunostained as described previously (Rognoni *et al*, 2016). Confocal microscopy was performed with a Nikon A1 confocal microscope using 10× or 20× objectives. Image processing was performed with Nikon ND2 Viewer Software, ImageJ (Fiji), Photoshop CS6 (Adobe) and Icy (version 1.6.0.0) software.

### Collagen gel assay

To study the proliferation of neonatal fibroblasts inside 3D extracellular matrix, we used double-layered collagen gels to avoid the unwanted attachment of fibroblasts to the tissue culture plastic. For the lower gels, rat tail collagen (1 mg/ml, Corning) in acetic acid was neutralised with NaOH and mixed with MQ-H₂0, HEPES

(25 mM), 10× MEM (1×), foetal bovine serum (15%) and L-glutamine (2 mM) on ice. Concentrations above represent the final values. The collagen mixture was pipetted into 96-well plates (50 µl per well) and incubated for 45–60 min (37°C, 5% $CO_2$) until a solid gel had formed. Mouse neonatal fibroblasts were added to the upper layer (1,000 cells per well) that was prepared in the same way as the lower gel, and the cells were mixed with the collagen mixture on ice. 100 µl of MEM (supplemented with 15% FBS, L-glutamine and penicillin-streptomycin) was added after the second layer of gel had formed. Alternatively, cells were plated on collagen-coated tissue culture wells (20 µg/ml in PBS, 1 h, 37°C, 5% $CO_2$).

For the measurement of cell proliferation, medium was removed and 100 µl of collagenase I from *Clostridium histolyticum* (2.75 µg/ml, Gibco, in the culture medium) was added. The final concentration of collagenase was re-adjusted with the culture medium in the case of the wells without collagen gels. 50 µl of CellTiter-Glo reagent (Promega) was added after 1 h (37°C, 5% $CO_2$) digestion. Wells were mixed vigorously and luminescence was measured by a plate reader (GloMax Discover, Promega).

To monitor proliferation capacity of quiescent neonatal fibroblasts on release from 3D ECM, cells were kept in collagen gels for 72 h and released by collagenase digestion as described above. Cells were plated on plastic, and proliferation was measured by Incucyte (Sartorius). Fold increase in cell confluency 24 h after plating was plotted over a period of 63 h.

### De-epidermised dermis (DED) organotypic culture

Primary human fibroblasts were isolated as described previously (Philippeos *et al*, 2018). Human adult dermal fibroblasts were cultured in Dulbecco's modified Eagle medium (DMEM) + 10% (v/v) FBS, 2 mM L-glutamine, 100 U/ml penicillin and 100 µg/ml streptomycin (Gibco). Culture flasks were incubated at 37°C in a humidified atmosphere with 5% $CO_2$ and passaged every 3–5 days, when 80% confluent. Cells were used between passages 1–6 for all studies. Stock cultures of primary normal human keratinocytes (NHKs, strain km) were obtained from surgically discarded foreskin and grown on 3T3-J2 feeder cells. NHKs were used for DED experiments between passages 2–5. 3T3-J2 fibroblasts were originally obtained from Dr James Rheinwald (Department of Dermatology, Skin Disease Research Center, Brigham and Women's Hospital, Boston, MA), not authenticated. All cell stocks were routinely tested for mycoplasma contamination and were negative. NHKs were cultured in complete FAD medium, containing 1 part Ham's F12, 3 parts DMEM, $10^{-4}$ M adenine, 10% (v/v) FBS, 0.5 µg/ml hydrocortisone, 5 µg/ml insulin, $10^{-10}$ M cholera toxin and 10 ng/ml EGF, on mitotically inactivated 3T3-J2 cells as described previously (Rheinwald & Green, 1975; Jones & Watt, 1993).

De-epidermised dermis (DED) was prepared as described previously (Rikimaru *et al*, 1997). Briefly, adult human skin was divided into 1–2 $cm^2$, heated at 52°C for 20 min and the epidermis separated from the dermis with forceps. The dermis was depleted of cells by at least 10 freeze–thaw cycles and irradiated once with 60 Gy. Before fibroblasts were seeded onto DEDs, the tissue was placed into 6-well hanging cell culture inserts (Millipore) and equilibrated with DMEM. $5 \times 10^5$ primary fibroblasts were injected into each DED using U-100 insulin syringes (BD) from the epidermis surface. The DEDs were then incubated for 72 h completely submerged in

DMEM. Medium was changed to FAD medium with an air–liquid interface, and $1 \times 10^6$ keratinocytes were seeded on top of the DED. DEDs were maintained in culture with FAD medium at an air–liquid interface for 3 weeks with media changes every 48 h. For digestion, 20–30 µl of digestion enzymes from a whole-skin dissociation kit (Miltenyi) were injected into the DED using U-100 insulin syringes, and incubated at 37°C for 3 h, without any medium. DEDs were then fully submerged in complete DMEM for 1 h to wash out digestion enzymes, and the air–liquid interface was re-established with FAD medium for 24–96 h.

### Collagen analysis

For collagen quantification, 12 µm cryosections from mouse back skin were stained with Picrosirius red using a standard method (Lattouf *et al*, 2014). Briefly, the sections were fixed (7 min, 4% PFA), washed twice with water and stained for 1 h in Picrosirius red solution [0.1% Sirius red F3B (Sigma) in saturated aqueous solution of picric acid]. After staining, sections were washed twice with acidified water (0.5% acetic acid), dehydrated, cleared with xylene and mounted with DPX mounting medium (Sigma). The images were taken with a Zeiss Axiophot microscope and AxioCam HRc camera under plane polarised light that shows the collagen fibres as green, orange and yellow against a black background. The intensity of light was adjusted to give a linear response for quantification. The quantification of total collagen fibres was performed using Fiji imaging software. The collagen pixels were selected with the colour threshold tool (hue 0–100, saturation 0–255 and brightness 230–255), and the binary images were created based on the selection. Collagen density was determined by dividing the area of collagen pixels by total tissue area. Total tissue area was manually cropped from the basement membrane to fat cells excluding hair follicles.

For collagen hybridising peptide (CHP) staining (Hwang *et al*, 2017; Li & Yu, 2013), 12 µm cryosections of back skin were fixed with 4% paraformaldehyde/PBS (10 min at room temperature), permeabilised with 0.1% Triton X-100/PBS (10 min at room temperature), blocked with 5% BSA/PBS (1 h at room temperature) and stained with the indicated primary antibodies and 5 µM B-CHP (BIO300, 3Helix) overnight at 4°C. According to the manufacturer's instructions, the B-CHP probe was heated for 5 min at 80°C before adding it to the primary antibody mix, which was immediately applied to the tissue sections. Sections were washed three times with PBS and incubated with appropriate secondary antibody and streptavidin–AlexaFluor647 (S32357, Thermo Fisher) for 1 h at room temperature. After washing the sections with PBS and incubating them with DAPI (1 µg/ml diluted 1:50,000) for 10 min at room temperature, the samples were mounted with ProLong® Gold Antifade Mountant (Thermo Fisher). Confocal microscopy was performed with a Nikon A1 confocal microscope using 10× or 20× objectives.

### Transmission electron microscopy (TEM)

Transmission electron microscopy of mouse dorsal skin was performed by the Centre for Ultrastructural Imaging at King's College London following fixation in 4% paraformaldehyde/PBS for 1 h. TEM (JOEL JEM) was used to obtain images from upper and

lower dermis. Using ImageJ (Fiji), 250 × 250 nm random areas were selected and collagen fibre diameter and collagen fibre density were measured.

### Flow cytometry and cell cycle analysis

Dermal fibroblasts were isolated at indicated time points, as previously described (Collins *et al*, 2011; Jensen *et al*, 2010). Briefly, back skin was incubated for 1 h in dispase–trypsin solution at 37°C, after which the epidermis was separated from the dermis and discarded. The dermis of neonatal mice was minced and incubated in 0.25% collagenase in FAD medium at 37°C for 1 h. Adult dermis was minced and incubated in FAD medium containing 1.25 mg/ml collagenase type I (Invitrogen), 0.5 mg/ml collagenase type II (Worthington), 0.5 mg/ml collagenase type IV (Sigma), 0.1 mg/ml hyaluronidase IVS (Sigma) and 50 U/ml DNase I at 37°C for 1 h. Enzyme activity was neutralised by the addition of serum-containing medium.

The dermal cell suspension was passed through a 70 μm cell strainer, washed twice with PBS and labelled with the following antibodies according to standard procedures: CD26 PercPcy5.5 (eBioscience, 45-0261), Ly-6A/E PE (eBioscience, Clone D7), CD45 APC (eBioscience, Clone), CD31 APC (eBioscience, Clone D7), CD234 APC (eBioscience, Clone D7) and Dlk1 PE (MBL International, D187-5). DAPI was used to exclude dead cells. Labelled cells were sorted on a BD FACSAria Fusion. For cell cycle analysis, isolated cells were fixed with 70% EtOH for 1 h at 4°C, washed with PBS and treated with 250 μl RNAse A (Sigma; 100 μg/ml in PBS) for 15 min at room temperature. Prior to FACS analysis with a BD FACSCanto II, cells were incubated with 25 μl propidium iodide (Sigma; 500 μg/ml stock solution) for 10 min at room temperature, and cells were gated for GFP$^+$ expression. Data analysis was performed using FlowJo software version 7.6.5 (Tree Star, Ashland, OR).

### Microarray analysis

We reanalysed the previously published microarrays GEO GSE32966 (Neonatal (P2) and P50 back skin PDGFRαH2BEGFP fibroblasts; Collins *et al*, 2011).

### *In vivo* live imaging

*In vivo* live imaging was performed as described in Hiratsuka *et al* (2015). In brief, two-photon excitation microscopy was performed with an A1RMP upright microscope, equipped with a 25×/1.10 water-immersion objective lens (CF175 Apo LWD 25XW Nikon) and a Ti:Sapphire laser (0.95 W at 900 nm; Coherent Chameleon II laser). The laser power used for observation was 2–10%. Scan speed was 4 μs/pixel. The excitation wavelength was 770 nm for second harmonic generation. *Z*-stack images were acquired with a view field of 0.257 mm² in 5 μm steps, and for whole wound maximum projections, images were stitched together.

For adult mouse back skin imaging, the fur was clipped and hair follicles were removed with depilation cream 1 h before wounding or 24 h before imaging unwounded skin. Imaging was started at the time points indicated. Throughout imaging, mice were anesthetised by inhalation of vaporised 1.5% isoflurane (Cp-Pharma) and placed in the prone position in a heated chamber maintained at 37°C. The

back skin was stabilised between a cover glass and a thermal conductive soft silicon sheet as previously described (Hiratsuka *et al*, 2015). The wound healing process was imaged at the indicated days after wounding. Time-lapse images were acquired every 30 min. A total of two mice per time point were examined, and the duration of time-lapse imaging was 7 ± 4 h per mouse. Time-lapse imaging was aborted if the body temperature, breathing condition or hydration of a mouse deteriorated. Optimisation of image acquisition was performed to avoid fluorescence bleaching and tissue damage and to obtain the best spatiotemporal resolution. Acquired images were analysed with MetaMorph (Universal Imaging, West Chester, PA) and Fiji imaging software (ImageJ, NIH). The fibroblasts of the PDGFRαH2BEGFP transgenic mice can be readily detected because of nuclear GFP expression (Collins *et al*, 2011) and the fibrillar collagen of the dermis is autofluorescent, enabling ready visualisation.

### Mathematical modelling

#### *Calculation of cell division rate*
The calculation of cell division rate has been described previously (Rognoni *et al*, 2016). Briefly, height, length and dermis diameter were measured (*n* = 3 mice per time point and gender), and the dermis volume was estimated by representing the mouse trunk as a cylinder. Cell densities were obtained from Fig 1A and B, and the cell numbers at E17.5 ($N_E$), P2 ($N_N$) and P50 ($N_A$) were estimated by multiplying cell density and dermis volume. The predicted cell division rate is calculated by the $\log_2$ of the $N_{older}/N_{younger}$ ratio.

#### *1D ODE models*
The negative feedback loop between proliferating fibroblasts and ECM was implemented in Mathematica.

#### *Dermis maturation*
We set out to mechanistically understand the process whereby fibroblasts orchestrate dermal maturation and wound healing. It is well established that during embryonic developmental stages (E), mouse fibroblasts proliferate and this process is driven by the presence of growth factors (Whitman & Melton, 1989), which here we designate by *GF*. Driskell *et al* (2013) showed that around E16.5 the multi-potent fibroblasts start maturing, giving rise to the two fibroblasts lineages, papillary and reticular. As both papillary and reticular fibroblasts are proliferative cells, for modelling simplification purposes, we combined the two lineages in a global proliferating fibroblasts (*PF*) term. During development, *PF* are progressively arrested at the G1 phase of the cell cycle and, in our model, we designate these arrested cells by quiescent fibroblasts (*QF*). While both *PF* and *QF* are metabolically active, *QF* are more efficient at depositing extracellular matrix (*ECM*) and *PF*s' main function is proliferation. This difference is the basis of the essential architecture of the dermis. As observed in Fig 1D, the increase in the amount of *ECM* is accompanied by a decrease in the number of *PF*, suggesting that *ECM* has a repressive effect on *PF*. We therefore assume that the reduction in *PF* by the presence of *ECM* can be described by a negative feedback mechanism of the following form:

$$PF + ECM \rightarrow QF + ECM \tag{1}$$

This is a catalytic mechanism, in which the presence of *ECM* enhances the entry into quiescence of proliferating cells via a cell-to-matrix interaction (Dilão & Muraro, 2010). All the processes described are summarised in the regulatory diagram shown in Fig 3A, and the kinetic diagrams describing the regulatory diagram are as follows:

$$
\begin{cases}
GF + PF \xrightarrow{k_1} 2PF \\
PF \underset{k_{-2}}{\overset{k_2}{\rightleftharpoons}} QF \\
QF \xrightarrow{k_3} QF + ECM \\
\boxed{PF + ECM \xrightarrow{k_4} QF + ECM} \\
PF \xrightarrow{k_5} \varnothing \\
QF \xrightarrow{k_6} \varnothing \\
ECM \xrightarrow{k_7} \varnothing
\end{cases}
\tag{2}
$$

As usual, rate constants $k_i$ are non-negative. The box in system (2) highlights the mechanism whereby the presence of *ECM* manifests as a negative feedback loop, promoting the transition between *PF* and *QF*. We will first describe the behaviour of *PF* and *ECM* during whole dermal maturation. We start by assuming that during the first stage of embryonic development the growth factor *GF* is not a limiting factor. The mechanism outlined by the kinetic diagrams (2) is described by the following system of equations:

$$
\begin{cases}
\dfrac{dPF}{dt} = k_1 GF\,PF - k_2 PF + k_{-2} QF - k_4 ECM\,PF - k_5 PF \\
\dfrac{dQF}{dt} = k_2 PF - k_{-2} QF + k_4 ECM\,PF - k_6 QF \\
\dfrac{dECM}{dt} = k_3 QF - k_7 ECM
\end{cases}
\tag{3}
$$

For simplification, *PF*, *QF* and *ECM* denote amounts of either cells or proteins. With the new constant

$$
\beta = k_1 GF - k_2 - k_5,
\tag{4}
$$

the system of equation (3) simplifies to

$$
\begin{cases}
\dfrac{dPF}{dt} = \beta PF + k_{-2} QF - k_4 ECM\,PF \\
\dfrac{dQF}{dt} = k_2 PF - k_{-2} QF + k_4 ECM\,PF - k_6 QF \\
\dfrac{dECM}{dt} = k_3 QF - k_7 ECM
\end{cases}
\tag{5}
$$

The system of equation (5) is our proposed mathematical model for the maturation process of the dermis.

The system of equation (5) has a fixed point other than the trivial solution:

$$
\begin{cases}
PF^* = \dfrac{k_7(k_{-2}(\beta + k_2) + \beta k_6)}{k_3 k_4 (\beta + k_2)} \\[2ex]
QF^* = \dfrac{k_7(k_{-2}(\beta + k_2) + \beta k_6)}{k_3 k_4 k_6} \\[2ex]
ECM^* = \dfrac{k_{-2}(\beta + k_2) + \beta k_6}{k_4 k_6}
\end{cases}
\tag{6}
$$

From equation (5), it follows that a model without the proposed feedback mechanism ($k_4 = 0$) would lead to a zero asymptotic concentration of *PF*, *QF* and *ECM* ($PF^* = 0$, $QF^* = 0$, $ECM^* = 0$), not compatible with the data. The stability of the steady state (6) can be analysed through the Jacobian matrix of (5), calculated at the fixed point (6). The Jacobian matrix is given by:

$$
\mathcal{J} = \begin{pmatrix}
\beta - \frac{k_{-2}(\beta+k_2)+\beta k_6}{k_6} & k_{-2} & -\frac{(k_{-2}(\beta+k_2)+\beta k_6)k_7}{(\beta+k_2)k_3} \\
k_2 + \frac{k_{-2}(\beta+k_2)+\beta k_6}{k_6} & -k_{-2} - k_6 & \frac{(k_{-2}(\beta+k_2)+\beta k_6)k_7}{(\beta+k_2)k_3} \\
0 & k_3 & -k_7
\end{pmatrix}
\tag{7}
$$

If all the eigenvalues of the Jacobian matrix (7) have negative real parts, the fixed point (6) is stable, leading to a steady dermis structure. We will use this to constrain the fitting of the model parameters to the experimental data, as our goal now was to obtain the parameters that offer the best fit. The kinetic model described by the system of equations (5) and by the regulatory mechanism depicted in (2) depends on seven independent parameters, with different ranges, and three initial conditions.

We have two independent experimental data sets that we must fit. Table 1 contains the quantification of the proportion of proliferating fibroblasts, as a percentage of the total number of fibroblasts, during mouse development. Table 2 contains the quantification of the total amount of collagen, as measured by Picrosirius red staining, with age.

To fit the experimental data in Tables 1 and 2 with the model equation (5), and due to the large number of free parameters, we follow a Monte Carlo technique adapted to the particularity of having two data sets to fit. In this multi-objective optimisation problem, we are in the context of a Pareto optimisation problem, which implies that there are no unique solutions (Dilão *et al*, 2009; Muraro & Dilão, 2013). To implement the Pareto optimisation procedure, we calculated the mean square deviations of the two data sets, for each set of parameter values, and chose a solution on the Pareto front of the multi-objective optimisation problem. Any solution belonging to the Pareto front is a minimal Pareto solution, in the sense that the value of one objective cannot be improved without degrading the value of the other objective (Dilão *et al*, 2009; Muraro & Dilão, 2013). In this case, the objectives are the minimum of the mean square deviations.

The fit procedure starts with a random choice of the seven parameters in model equation (5) and the random choice of the initial

**Table 1. Temporal changes in fibroblast proliferation.**

| Time (days) | PF (%) | Standard deviation |
|---|---|---|
| 10.5 | 75.1 | 1.4 |
| 16.5 | 46.5 | 6.4 |
| 17.5 | 52.8 | 2.8 |
| 18.5 | 51.0 | 2.0 |
| 19.5 | 30.8 | 4.8 |
| 21.5 | 13.5 | 1.6 |
| 29.5 | 2.3 | 1.4 |
| 41.5 | 1.5 | 0.4 |
| 59.5 | 0.69 | 0.1 |
| 69.5 | 0.44 | 0.04 |

**Table 2. Experimental data for ECM.**

| Time (days) | ECM (%) | Standard deviation |
|---|---|---|
| 10.5 | 0 | 0 |
| 16.5 | 0.3 | 0.2 |
| 17.5 | 0.4 | 0.4 |
| 18.5 | 12.1 | 1.4 |
| 19.5 | 7.9 | 0.8 |
| 21.5 | 17.4 | 1.9 |
| 24.5 | 37.8 | 1.1 |
| 29.5 | 45.4 | 2.4 |
| 31.5 | 46.5 | 3.3 |
| 33.5 | 57.2 | 4.5 |
| 35.5 | 60.4 | 3.9 |
| 38.5 | 64.7 | 4.3 |
| 40.5 | 72.0 | 5.6 |
| 73.5 | 88.3 | 5.1 |

conditions $PF(t = 10.5) = PF_0$ and $QF(t = 10.5) = QF_0$. The initial condition for $ECM$ is assumed to be zero, $ECM(t = 10.5) = 0$. Then, for this choice of parameters, we test if the steady state (6) is stable. If the real parts of all the eigenvalues of the Jacobian matrix (7) are negative, we accept these parameters as an eventual solution of the optimisation problem. For each random choice of acceptable parameters and initial conditions, we calculate the mean square deviations from the $PF$ and $ECM$ data sets. After iterating this procedure 100,000 times, for random choices of the parameter values, we calculate the solutions on the Pareto front obtained from the mean square deviation pairs. Then, one solution on the Pareto front is chosen.

The parameter values and initial conditions that fit the experimental data in Tables 1 and 2 are depicted in Table 3. In Fig 3C, we show the experimental data and the solutions obtained with model equation (5) with the parameters in Table 3. The immediate conclusion is that the negative feedback loop resulting from the catalytic model summarised in Fig 3A and described by equation (5) describes with high accuracy the mechanism of dermis maturation.

**Table 3. Parameter values and initial conditions for model equation (5) that fit the experimental data in Tables 1 and 2.**

| Parameters | Fitted values |
|---|---|
| $\beta$ | 0.3010 |
| $k_2$ | 1.0102 |
| $k_{-2}$ | 0.6084 |
| $k_3$ | 0.0610 |
| $k_4$ | 0.2008 |
| $k_6$ | 0.0424 |
| $k_7$ | 0.8380 |
| $PF_0$ | 0.3530 |
| $QF_0$ | 0.1166 |

### Wound healing

It is well established that upon wounding dermal fibroblasts become activated and start proliferating (Rognoni et al, 2016). As our model proposes a negative feedback mechanism describing the interaction between $PF$ and $ECM$, we decided to investigate the dynamic mechanism of regeneration when the local steady state is perturbed, as in the case of a wound for example.

In our model, the system of equations (5) describes the local concentration of cells in the dermis during maturation and homeostasis, assuming that the structure of the dermis is spatially homogeneous. To describe the wound healing mechanism, we assume that after wounding the regeneration process is initiated with the motility of fibroblasts at the boundary of the wound. The closer to the wound, the more the system is away from the steady state (6). The simplest mechanism to describe regeneration is to assume that at the wound region there is movement of $PF$ and $QF$ cells, dynamically regenerating the structure of the dermis. On the other hand, as $ECM$ is a mixture of proteins, it is also plausible to assume that it does not move, in the sense that it is deposited near quiescent fibroblasts. In this case, the wound healing mechanism is only associated with the motility of the $PF$ and $QF$ cells. Under these conditions, the spatial structure of the dermis is described by the partial differential system of equations

$$\begin{cases} \dfrac{\mathrm{d}PF}{\mathrm{d}t} = \beta PF + k_{-2}QF - k_4 ECM\,PF + D\dfrac{\partial^2 PF}{\partial x^2} \\ \dfrac{\mathrm{d}QF}{\mathrm{d}t} = k_2 PF - k_{-2}QF + k_4 ECM\,PF - k_6 QF + D\dfrac{\partial^2 QF}{\partial x^2} \\ \dfrac{\mathrm{d}ECM}{\mathrm{d}t} = k_3 QF - k_7 ECM \end{cases} \quad (8)$$

where the constant $D$ is a diffusion coefficient, equal for both $PF$ and $QF$ cells. To simplify, we have considered that dermis is developed along a one-dimensional domain, with periodic boundary conditions (circular domain). Due to the simplicity of our assumptions, we can only test qualitatively the results of the model. We will not consider the case of periodic boundary conditions, and a more detailed mathematical analysis of the model equations will be developed in a subsequent publication.

To be more specific, we consider the one-dimensional spatial domain of length $L = 1$. This spatial domain represents the dermis. We also assume that the parameter values in Table 3 still hold. The situation we want to analyse is the mechanism driving the changes in the cell density inside the wound. For that, we consider that a normal dermis is characterised by a uniform concentration of fibroblasts and extracellular matrix in steady state. To test the model, we further consider a wound in the spatial region $x \in [0.45, 0.5]$. In this case, the wounded tissue is represented by the initial conditions in (9):

$$\begin{cases} PF(x, t = 0) = PF^* & \text{for } x \in [0, 0.45] \text{ and } x \in [0.5, 1], \\ & \text{and } PF(x, t = 0) = 0, \text{ otherwise} \\ QF(x, t = 0) = QF^* & \text{for } x \in [0, 0.45] \text{ and } x \in [0.5, 1], \\ & \text{and } QF(x, t = 0) = 0, \text{ otherwise} \\ ECM(x, t = 0) = ECM^* & \text{for } x \in [0, 0.45] \text{ and } x \in [0.5, 1], \\ & \text{and } ECM(x, t = 0) = 0, \text{ otherwise} \end{cases} \quad (9)$$

In this formulation $PF$, $QF$ and $ECM$ are densities. The initial conditions (9) are not a steady-state solution of the partial

differential equation (8), and therefore the local densities evolve with time.

To integrate the partial differential equation system (8) from the initial conditions (9), we used a calibrated numerical method for reaction-diffusion equations (Dilão & Sainhas, 2011), and we chose a diffusion coefficient in such a way that the percentage of fibroblasts is attained by day 4.5 after wounding, in agreement with the experimental results of Fig 6A. After several numerical tests, we estimated the diffusion coefficient to be $D = 2.10^{-4}$ cm$^2$/day. We show in Fig 6D the temporal evolution of the percentage of fibroblasts and extracellular matrix inside the wound, where we can observe that the results are qualitatively similar to the experimental data in Fig 6A.

The values of fibroblasts and extracellular matrix inside the wound were calculated by:

$$
\begin{cases}
\overline{PF}(t) = \int\limits_{0.45}^{0.5} PF(x,t)\,\mathrm{d}x \\[2ex]
\overline{QF}(t) = \int\limits_{0.45}^{0.5} QF(x,t)\,\mathrm{d}x \\[2ex]
\overline{ECM}(t) = \int\limits_{0.45}^{0.5} ECM(x,t)\,\mathrm{d}x \\[2ex]
\overline{\overline{PF}}(t) = 100\,\dfrac{\overline{PF}(t)}{\overline{PF}(t) + \overline{QF}(t)}
\end{cases} \tag{10}
$$

where *PF(x,t)*, *QF(x,t)* and *ECM(x,t)* are the numerical solutions of equation (10).

### Tissue model: Glazier–Graner–Hogeweg/Cellular Potts simulations

The Glazier–Graner–Hogeweg/Cellular Potts Model (GGH/CPM) represents a single cell as an extended domain of sites (pixels) on a regular lattice that share a common index σ. For detailed information about GGH/CPM, see Belmonte *et al* (2016) and Swat *et al* (2012). In brief, a cellular Potts model was implemented using the CompuCell3D platform (Swat *et al*, 2012). Cells in this model are represented as connected domains on a grid with the cell-free area represented as a special domain. Chemoattractant concentrations are represented on the same grid. Cell dynamics result from a series of attempts to expand the domains at randomly selected grid sites. Whether an expansion attempt is accepted or not depends on a set of rules, which thus determine cell dynamics. The time unit is expressed as Monte Carlo step (MCS), defined as the number of attempted site switches equal to the number of grid sites in the simulation. Our CC3D model computer code is available on GitHub: https://github.com/aopisco/DermisMaturation. Simulations represent a 3D section of mouse skin. Built-in routines were used for the implementation of chemotaxis and secretion. Fibroblasts are initialised as 10 cells disposed radially in contact with the epidermis (wall boundary conditions) and surrounding the body cavity (lumen). Lumen and epidermal integrity, cell divisions, cell differentiation, cell fate conversion and wound healing were implemented as custom routines.

Dermal maturation model assumptions:

- Epidermal gradient: In line with the literature (Collins *et al*, 2011; Lichtenberger *et al*, 2016), we assumed that fibroblast proliferation relies on an epidermal signal gradient, which decays over time. Because of that, we implemented the *GrowthSteppable* routine, which reads the strength of the epidermal gradient and determines the growth of the cells based on that.
- Transition to quiescence: Our hypothesis is that the physical presence of ECM triggers the fibroblasts to stop proliferating. This is implemented in the *TransitionSteppable* routine by switching all fibroblasts from proliferating to quiescent when more than 50% of the cell surface area is in contact with ECM.
- The parameters were manually adjusted such that MCS = 300 coincided with the *in vivo* proliferation data measured at P0, and no parameter sensitivity analysis was performed.

Wound healing model assumptions:

- Blood clot and immune clot structures: In order to fill the space of the cut, we introduced two new structures. The blood clot is a solo illustrative structure, helping to visualise skin repair dynamics. The immune clot is an active structure, responsible for producing the wound healing signal.
- Wound healing gradient: Based on the literature (Eming *et al*, 2014; Shaw & Martin, 2016), we introduced a wound healing chemical field, which is in essence identical to the epidermal gradient but cannot diffuse far from the producing cells. We hypothesise that this gradient is able to push quiescent fibroblasts back into proliferation, that is, in response to the wound healing signal quiescent fibroblasts become myofibroblasts.
- Epidermal growth: Based on Park *et al* (2017), we included epidermal growth as one of the first steps of tissue repair. This is an important component of the model as it (i) recapitulates the macroscopic *in vivo* behaviour, and (ii) aids the healing process by creating a localised source of epidermal signalling around the wound bed that promotes fibroblast proliferation.
- We assume that the myofibroblasts behave in a similar fashion to the proliferating fibroblasts. Therefore, in our model, both cell types divide and deposit ECM at the same rates.

### Quantification and statistical analysis

Data analysis was performed with GraphPad Prism 7 software. Unless stated otherwise, data are means ± standard deviation (s.d.). The spot detector plugin of Icy software (version 1.6.0.0) was used for unbiased identification and quantification of cells labelled with RFP, CFP or CFP (Confetti lineage tracing) and cell nuclei labelled with DAPI, Ki67, R26Fucci2a or PDGFRαH2BEGFP. Cell densities and lineage traced cells were quantified from at least eight 12 or 60 μm horizontal skin wholemounts or wound bed sections per mouse.

## Data availability

The data sets and computer code produced in this study are available in the following databases:

- Microarray data: Gene Expression Omnibus GSE32966 (https://www.ncbi.nlm.nih.gov/geo/query/acc.cgi?acc=GSE32966)

- 3D model of Dermis Maturation and Wound Healing: CompuCel-l3D code (https://github.com/aopisco/DermisMaturation)

**Expanded View** for this article is available online.

## Acknowledgements

FMW gratefully acknowledges financial support from the Medical Research Council (MR/PO18823/1) and Wellcome Trust (206439/Z/17/Z). ER is the recipient of an EMBO long-term fellowship (aALTF 594-2014) and an EMBO Advanced Fellowship (aALTF 523-2017); TH is the recipient of a Marie Skłodowska Curie Fellowship (H2020-MSCA-IF-2015_704587); and KHS is the recipient of a Finnish Cultural Foundation Fellowship. We are grateful to Yang Li (University of Utah) for providing collagen hybridising peptide and Gema Vizcay-Barrena of the KCL Centre for Ultrastructural Imaging for technical support. We thank the Nikon Imaging Centre and BSU staff at KCL for expert assistance. We also acknowledge the use of Core Facilities provided by the generous financial support from the Department of Health via the National Institute for Health Research (NIHR) comprehensive Biomedical Research Centre award to Guy's & St Thomas' NHS Foundation Trust in partnership with King's College London and King's College Hospital NHS Foundation Trust. We would like to thank Prof. James Glazier (Indiana University) for the training provided at the CompuCell3D 2017 workshop.

## Author contributions

ER and AOP conceptualised the study, designed the experiments, delineated the mathematical modelling and analysed the results. ER performed *in vivo* experiments. AOP performed mathematical modelling and bioinformatics analysis. TH performed the *in vivo* live imaging. KHS performed the Picrosirius red stainings and quantifications and collagen gel experiments. JMB assisted with the CompuCell3D modelling. SAM performed TEM imaging and quantification. CP performed DED organotypic cultures. RD assisted with the model design and implementation of the 1D simulations. ER and AOP wrote the manuscript and produced the Figs. FMW oversaw the study and co-wrote the manuscript.

## Conflict of interest

The authors declare that they have no conflict of interest. FMW is currently on secondment as Executive Chair of the Medical Research Council.

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
