## [Review Process File · Molecular Systems Biology]

Fibroblast state switching orchestrates dermal maturation and wound healing

Emanuel Rognoni, Angela Oliveira Pisco, Toru Hiratsuka, Kalle Sipilä, Julio M. Belmonte, Seyedeh Atefeh Mobasser, Christina Philippeos, Rui Dilão and Fiona M. Watt.

Review timeline:	Submission date:	18 th December 2017
	Editorial Decision:	23 rd February 2018
	Revision received:	11 th June 2018
	Editorial Decision:	10 th July 2018
	Revision received:	2 nd August 2018
	Accepted:	3 rd August 2018

Editor: Maria Polychronidou

Transaction Report:

1st Editorial Decision

23rd February 2018

Thank you again for submitting your work to Molecular Systems Biology. We have now heard back from the two referees who agreed to evaluate your study. As you will see below, the reviewers think that the presented findings seem interesting. They raise however a series of concerns, which we would ask you to address in a revision of the manuscript.

The reviewers' recommendations are rather clear and therefore I think that there is no need to repeat all the points listed below. One particularly important point raised by both reviewers refers to the need to provide further evidence supporting the causal relationship between the ECM and cell proliferation. Please let me know in case you would like to discuss further any of the issues raised by the reviewers.

REFeree REPORTS

Reviewer #1:

Summary

This paper analyzes the role of fibroblasts and extracellular matrix in the development of homeostasis and the response to wounding in mammalian skin. Motivated by experiments in embryonic and neonatal mice that show an inverse correlation between fibroblast proliferation and extracellular matrix (ECM) density and that quiescent fibroblasts are more efficient at producing ECM than those that proliferate, the authors hypothesized that the ECM negatively regulates fibroblast proliferation and this regulation plays a key role in determining the structure of the dermis.

To elucidate this process, the authors develop several mathematical models that range from well-mixed (spatially homogeneous) models to 3D models of the dermis that account for spatiotemporal interactions of multiple cell types. The models suggest that the negative feedback hypothesis is sufficient to achieve dermal homeostasis. The models suggest that active cell migration is not needed for maturation of the dermis (developmental approach to homeostasis), but that active migration is an important component of wound healing. Live imaging in mice confirms these conclusions.

General remarks

How tissues reach their correct size with the right distribution and numbers of cells and how these tissues regenerate after perturbations are fundamental problems in Developmental Biology that are still not well understood. This is an interesting study that tackles these questions for the dermal layer of mammalian skin and introduces what appears to be a new idea—a negative feedback loop between ECM and fibroblasts—for control of the dermal structure. The idea that active motion of fibroblasts is not needed for maturation is also interesting. It is to be expected that active motion is needed to heal wounds, however. The paper should be of interest to Developmental Biologists in general and Skin Biologists in particular and to Systems Biologists who seek to understand problems in Development through theoretical and computational methodologies.

Major Points

The data presented convincingly show that there is a correlation between ECM deposition and fibroblast proliferation. The mathematical models and the data suggest that this feedback is sufficient to characterize the approach to homeostasis during both maturation and wound healing. However, is the relationship between proliferation and ECM really causal? Could the authors provide evidence that it is the ECM that induces the anti-proliferative effect and not another signaling factor (e.g., produced by the fibroblasts themselves or other cells in the dermis) that induces quiescence? The result of this could be the same as the proposed mechanism but would provide a different interpretation of the results that would be important to know. The 3D tissue model that the authors use is based on the Cellular Potts model developed originally by Granier-Glazier-Hogeweg and implemented in the CompuCell3D framework. This is an appropriate modeling choice but unfortunately there are insufficient details presented either in the main text or the supplementary material for this reviewer to judge the simulations. In particular, there are numerous assumptions and it would be important to know which assumptions are critical to the authors' conclusions.

Minor Points

There are a number of typos in the paper that need to be corrected. The authors need to go over the manuscript carefully and also make sure all the text is consistent with the figures.

P6. Proliferative fibroblasts are "blue" in Fig EV3A.

P7. Epidermal signal strength was assumed to decline with age. Why? The authors should provide a reference

P9. Top paragraph. The figure references do not match the actual figure panels. (e.g., Fig 4I and 4G)

P20. Fig 1A caption. The references to the "left" and "right" panels are reversed.

P21. Fig 2 caption. "GF" is not defined.

Fig 2: What does % ECM density mean?

Fig EV1: Panel A. 1:4 should be 4:1

Fig EV5. Panel B. The text overlaps the region of interest

Reviewer #2:

The manuscript by Rognoni et al. describes a combined experimental and modeling work that focuses on understanding the spatiotemporal dynamics of dermis maturation during normal development and wound healing. The main premise of the manuscript is that the interrelation between proliferation and extracellular matrix (ECM) deposition underlies these processes. The

authors propose a model where fibroblast cells can switch between proliferating fibroblast (PF) and quiescent fibroblast (QF) states. They propose that the quiescent cells secrete ECM that inhibits the proliferating state and enhances the quiescent state. Experimentally, it is shown that proliferation is inversely proportional to ECM deposition both during normal maturation and during wound healing. It is also shown that during normal maturation there is no cell migration, but during wound healing there is a significant fibroblast migration from areas next to the wound.

The authors develop a mathematical model to describe the relation between proliferation, ECM and cell migration. They develop three types of models: (1) A non-spatial regulatory network model describing the feedback between PFs, QFs, and ECM. (2) A 1D reaction-diffusion type model that describes cellular dynamics of wound healing in a simplified 1D geometry. (3) A detailed cellular Potts model that describes both normal dermis maturation and wound healing process in 3D.

Overall the manuscript is well written, and the results provide compelling experimental and theoretical insights into the organization of the Dermis. It also fits quite well the scope of the MSB journal. However, there are a few important points that need to be addressed before it is accepted.

These include:

Major comments:

1. One of the main conjectures in the manuscript is that the ECM inhibits the proliferating state (PF) by enhancing the transition from that state to the quiescent state. The evidence for this feedback relies on the correlation between the increase of ECM and the decrease in proliferation. However, that correlation does not necessarily imply causation. It is important to provide evidence that inhibition of ECM deposition increases proliferation. It could be that the decline of proliferation is controlled by some other cellular process upstream of the ECM deposition.

2. Furthermore, on the modeling side, the authors should compare a mathematical model with a feedback of the ECM on PF (K4) to a model without such a feedback (but with some upstream signal that controls both decay in proliferation and ECM deposition). Are there unique features that can be explained only by the feedback model?

3. The authors argue that no active migration is required for the maturation process. The evidence to support that is a 200min movie (movie EV2) showing only small movement. This seems to me too short a movie to make this point as the maturation process proceeds over many days. I therefore think that a longer movie is required to make this point. Alternatively, the authors should quantitate the effective cell diffusion in their current movies and show that is consistent with no significant migration.

4. The authors use both a simplified 1D model (Fig. 5) and a 3D cellular Potts model to show wound closure. However, these models are fundamentally different in their assumptions. The 1D model simply assumes that the cell can diffuse into the wound. This seems an oversimplified assumption, as it well known that cells perform directional migration in response to signal from the wound. In fact, the author show that very nicely in movie EV6. This type of directional movement is inconsistent with random diffusion assumed in that model. I think that it would be much more insightful to introduce into the 1D model directed random walk.

5. The detailed 3D model shows that with the right assumptions, the wound closure can be recapitulated. However, the details are unclear. Information on the assumptions made and the details of the model are not provided. There is a general reference to previous work but this is not enough. In particular, it is not clear which features are required and which are not. For example, the detailed simulation (Fig. 6A) includes both blood clot and infiltrating immune cells. Are these two types required for the model to work? If so, under what conditions?

It also includes Epidermis proliferation in response to the wound healing. Is that also necessary? More generally, the model should provide insights into the essential processes sufficient for recapitulating the process and not only provide a nice movie that looks like the real behavior. It would be useful to perform some parameter sensitivity which would identify the essential parameters for the process.

6. As mentioned above, the authors should provide a more complete description of the 3D model and its parameters, as they have done for the 1D model.

Minor point:

1. The authors use the acronym MCS in multiple figure to denote time but do not define it anywhere.

Point-by-point reply to the reviewers**Reviewer #1:****Major Points**

The data presented convincingly show that there is a correlation between ECM deposition and fibroblast proliferation. The mathematical models and the data suggest that this feedback is sufficient to characterize the approach to homeostasis during both maturation and wound healing. However, is the relationship between proliferation and ECM really causal? Could the authors provide evidence that it is the ECM that induces the anti-proliferative effect and not another signaling factor (e.g., produced by the fibroblasts themselves or other cells in the dermis) that induces quiescence? The result of this could be the same as the proposed mechanism but would provide a different interpretation of the results that would be important to know.

We thank the reviewer for this important point, which was also raised by Reviewer 2. To obtain experimental evidence for a causal relationship between fibroblast proliferation and ECM deposition we have now included two new experimental approaches in a new figure (Fig.2). First, we show that fibroblasts reversibly stop proliferating when encased in a collagen gel. Secondly, we show that when dermal ECM is subjected to collagenase digestion fibroblasts resume proliferation. In both models the transition of fibroblasts between quiescence and proliferation occurs independently of an immune cell infiltrate. The new in vitro experiments demonstrate the strong interdependence of fibroblast proliferation and ECM deposition and strongly support our feedback model.

The 3D tissue model that the authors use is based on the Cellular Potts model developed originally by Granier-Glazier-Hogeweg and implemented in the CompuCell3D framework. This is an appropriate modeling choice but unfortunately there are insufficient details presented either in the main text or the supplementary material for this reviewer to judge the simulations. In particular, there are numerous assumptions and it would be important to know which assumptions are critical to the authors' conclusions.

We thank the reviewer for raising this point. We have now created a GitHub repository, where we have uploaded the 3D model (<https://github.com/aopisco/DermisMaturation>). Additionally the methods section has been expanded to provide the information that was lacking previously. Our work can now be easily reproduced and expanded, a critical feature for the systems biology/computational biology community.

Minor Points

There are a number of typos in the paper that need to be corrected. The authors need to go over the manuscript carefully and also make sure all the text is consistent with the figures.

We have now corrected these errors.

P6. Proliferative fibroblasts are "blue" in Fig EV3A.

This is now stated in the text.

P7. Epidermal signal strength was assumed to decline with age. Why? The authors should provide a reference.

We have now cited Lichtenberger et al., 2016 and Collins et al., 2011 in the text.

P9. Top paragraph. The figure references do not match the actual figure panels. (e.g., Fig 4I and 4G) P20.

This has now been corrected.

Fig 1A caption. The references to the "left" and "right" panels are reversed.

This has now been corrected.

P21. Fig 2 caption. "GF" is not defined.

This information is now added to the figure legend.

Fig 2: What does % ECM density mean?

We have replaced the Figure legend to make this clear.

Fig EV1: Panel A. 1:4 should be 4:1.

We have now changed this.

Fig EV5. Panel B. The text overlaps the region of interest.

We have moved the text outside of the figure.

Reviewer #2:

Major comments:

1. One of the main conjectures in the manuscript is that the ECM inhibits the proliferating state (PF) by enhancing the transition from that state to the quiescent state. The evidence for this feedback relies on the correlation between the increase of ECM and the decrease in proliferation. However, that correlation does not necessarily imply causation. It is important to provide evidence that inhibition of ECM deposition increases proliferation. It could be that the decline of proliferation is controlled by some other cellular process upstream of the ECM deposition.

We thank the reviewer for pointing this out, in common with Reviewer 1, who raised the same point. To obtain experimental evidence for a causal relationship between fibroblast proliferation and ECM deposition we have now included two new experimental approaches in a new figure (Fig.2). First, we show that fibroblasts reversibly stop proliferating when encased in a collagen gel. Secondly, we show that when dermal ECM is subjected to collagenase digestion fibroblasts resume proliferation. In both models the transition of fibroblasts between quiescence and proliferation occurs independently of an immune cell infiltrate. The new in vitro experiments demonstrate the strong interdependence of fibroblast proliferation and ECM deposition and strongly support our feedback model.

2. Furthermore, on the modeling side, the authors should compare a mathematical model with a feedback of the ECM on PF (K4) to a model without such a feedback (but with some upstream signal that controls both decay in proliferation and ECM deposition). Are there unique features that can be explained only by the feedback model?

We thank the reviewer for this comment. We have revised the Supplementary File EV1 (now part of Methods) by adding a sentence following equation (6), showing that the non-existence of a negative feedback loop would imply a zero-steady state for PF, QF and ECM, in contradiction with the data.

3. The authors argue that no active migration is required for the maturation process. The evidence to support that is a 200min movie (movie EV2) showing only small movement. This seems to me too short a movie to make this point as the maturation process proceeds over many days. I therefore think that a longer movie is required to make this point. Alternatively, the authors should quantitate the effective cell diffusion in their current movies and show that is consistent with no significant migration.

We agree and have now extended the movie to 700 min. We have also included stills from the later time points (Figure 3G).

4. The authors use both a simplified 1D model (Fig. 5) and a 3D cellular Potts model to show wound closure. However, these models are fundamentally different in their assumptions. The 1D model simply assumes that the cell can diffuse into the wound. This seems an oversimplified assumption, as it well known that cells perform directional migration in response to signal from the wound. In fact, the author show that very nicely in movie EV6. This type of directional movement is inconsistent with random diffusion assumed in that model. I think that it would be much more insightful to introduce into the 1D model directed random walk.

The motivation for using two different assumptions is that we wanted to show that with the simplest assumption, that is, diffusion, could macroscopically recapitulate the observed behaviour. Nevertheless, in the 3D tissue model, we wanted to describe the system more realistically and we can actually account for the directional migration that we observe by live imaging in wound healing.

5. The detailed 3D model shows that with the right assumptions, the wound closure can be recapitulated. However, the details are unclear. Information on the assumptions made and the details of the model are not provided. There is a general reference to previous work but this is not enough. In particular, it is not clear which features are required and which are not. For example, the detailed simulation (Fig. 6A) includes both blood clot and infiltrating immune cells. Are these two types required for the model to work? If so, under what conditions?

It also includes Epidermis proliferation in response to the wound healing. Is that also necessary? More generally, the model should provide insights into the essential processes sufficient for recapitulating the process and not only provide a nice movie that looks like the real behavior. It would be useful to perform some parameter sensitivity which would identify the essential parameters for the process.

We have now revised our methods section to include in detail all the model assumptions and their relevance for the model. Moreover, the code is now available in GitHub: <https://github.com/aopisco/DermisMaturation>. The new experiments in Figure 2A, B establish that the switch between fibroblast proliferation and quiescence can occur in the absence of epidermis, immune cells and a blood clot.

6. As mentioned above, the authors should provide a more complete description of the 3D model and its parameters, as they have done for the 1D model.

This is now provided in the method section.

Minor point:

1. The authors use the acronym MCS in multiple figure to denote time but do not define it anywhere.

We have now defined this in the text.

2nd Editorial Decision

10th July 2018

Thank you for sending us your revised manuscript. We have now heard back from reviewer #2 who was asked to evaluate your study. As you will see below the reviewer is satisfied with the modifications made and thinks that the study is now suitable for publication.

We are offering a "model curation service" together with Prof. Jacky Snoep and the FAIRDOM team. In brief, the aim is to enhance reproducibility and add value to papers including mathematical models. Jacky Snoep's summary on the model (*Model Curation Report*) is pasted below. As you will see, there are some minor issues, which we would ask you to fix when you submit your revision.

REFeree REPORTS.

Reviewer #2:

The authors have addressed all the points raised in the first round in an adequate manner. I particularly like the new experiments in Fig.2 showing a causal relation between ECM deposition and proliferation.

The description of the 3D model has also improved significantly.

I therefore recommend accepting the manuscript

****Model Curation Report****:

The authors sent me a Mathematica notebook with the models described in the MSB manuscript.

I could reproduce the simulation plots in the manuscript, and the model description in the manuscript is in agreement with the notebook.

I have only minor comments for improvement of Figure 3:

1) The schema in Fig. 3a is not consistent with the ODEs. I would advise to show all the rates in the schema, i.e. also include k_5 , k_6 , k_7 . In addition I would make a distinction between conversion (e.g. PF to QF) and activation (e.g. ECM effect on k_4). Now both types of reactions are shown as a solid arrow. I would suggest to use a dotted arrow for activation, simply to prevent confusion. Thus, show an additional solid line from PF to QF for k_4 , with a dotted arrow from ECM pointing at k_4 . Similarly, QF is not converted to ECM, but rather activates the synthesis, i.e. should be a dotted line.

2) If I am correct ECM in Fig. 3c is not a percentage, but a density, so it should not have units "(%)".

2nd Revision - authors' response

2nd August 2018

2ND Point-by-point reply to the reviewers

Model Curation Report

I have only minor comments for improvement of Figure 3:

1) The schema in Fig. 3a is not consistent with the ODEs. I would advise to show all the rates in the schema, i.e. also include k_5 , k_6 , k_7 . In addition I would make a distinction between conversion (e.g. PF to QF) and activation (e.g. ECM effect on k_4). Now both types of reactions are shown as a solid arrow. I would suggest to use a dotted arrow for activation, simply to prevent confusion. Thus, show an additional solid line from PF to QF for k_4 , with a dotted arrow from ECM pointing at k_4 . Similarly, QF is not converted to ECM, but rather activates the synthesis, i.e. should be a dotted line.

We have changed Fig 3A accordingly.

2) If I am correct ECM in Fig. 3c is not a percentage, but a density, so it should not have units "(%)".

We have changed the axis labelling accordingly.

Corresponding Author Name: Fiona M Watt
 Journal Submitted to: Molecular Systems Biology
 Manuscript Number: MSB-17-8174